# Differentially Private Statistical Inference through $\beta$-Divergence One Posterior Sampling

**Jack Jewson**[*]
Department of Economics and Business
Universitat Pompeu Fabra
Barcelona, Spain
`jack.jewson@upf.edu`

**Sahra Ghalebikesabi**[*]
Department of Statistics
University of Oxford
Oxford, UK
`sahra.ghalebikesabi@univ.ox.ac.uk`

**Chris Holmes**
The Alan Turing Institute
Department of Statistics
University of Oxford
Oxford, UK
`chris.holmes@stats.ox.ac.uk`

## Abstract

Differential privacy guarantees allow the results of a statistical analysis involving sensitive data to be released without compromising the privacy of any individual taking part. Achieving such guarantees generally requires the injection of noise, either directly into parameter estimates or into the estimation process. Instead of artificially introducing perturbations, sampling from Bayesian posterior distributions has been shown to be a special case of the exponential mechanism, producing consistent, and efficient private estimates without altering the data generative process. The application of current approaches has, however, been limited by their strong bounding assumptions which do not hold for basic models, such as simple linear regressors. To ameliorate this, we propose $\beta$D-Bayes, a posterior sampling scheme from a generalised posterior targeting the minimisation of the $\beta$-divergence between the model and the data generating process. This provides private estimation that is generally applicable without requiring changes to the underlying model and consistently learns the data generating parameter. We show that $\beta$D-Bayes produces more precise inference estimation for the same privacy guarantees, and further facilitates differentially private estimation via posterior sampling for complex classifiers and continuous regression models such as neural networks for the first time.

## 1  Introduction

Statistical and machine learning analyses are increasingly being done with sensitive information, such as personal user preferences [49], electronic health care records [72], or defence and national security data [27]. It thus becomes more and more important to ensure that data-centric algorithms do not leak information about their training data. Let $D = \{D_i\}_{i=1}^n = \{y_i, X_i\}_{i=1}^n \in \mathcal{D}$ denote a sensitive data set with $d$ dimensional features $X_i \in \mathcal{X} \subset \mathbb{R}^d$, and labels $y_i \subset \mathcal{Y} \subset \mathbb{R}$. Here, we are considering inference problems where a trusted data holder releases model parameters $\tilde{\theta}(D) \in \Theta$, that describe the relationship between $X$ and $y$, based on an arbitrary likelihood model $f(\cdot; \theta)$. Differential privacy

---

[*]Joint first authorship. Order decided based on coin flip.

37th Conference on Neural Information Processing Systems (NeurIPS 2023).

(DP) provides a popular framework to quantify the extent to which releasing $\tilde{\theta}(D)$ compromises the privacy of any single observation $D_i = \{y_i, X_i\} \in D$.

**Definition 1** (Differential Privacy, [26]). *Let $D$ and $D'$ be any two neighbouring data sets differing in at most one feature label pair. A randomised parameter estimator $\tilde{\theta}(D)$ is $(\epsilon, \delta)$-differentially private for $\epsilon > 0$ and $\delta \in [0, 1]$ if for all $\mathcal{A} \subseteq \Theta$, $P(\tilde{\theta}(D) \in \mathcal{A}) \leq \exp(\epsilon) P(\tilde{\theta}(D') \in \mathcal{A}) + \delta$.*

The privacy guarantee is controlled by the privacy budget $(\epsilon, \delta)$. While $\epsilon$ bounds the log-likelihood ratio of the estimator $\tilde{\theta}(D)$ for any two neighbouring data sets $D$ and $D'$, $\delta$ is the probability of outputs violating this bound and is thus typically chosen smaller than $1/n$ to prevent data leakage. DP estimation requires the parameter estimate to be random even after conditioning on the observed data. Thus, noise must be introduced into the learning or release procedure of deterministic empirical risk minimisers to provide DP guarantees.

The *sensitivity method* [26] is a popular privatisation technique in DP where the functional that depends on the sensitive data (i.e. a sufficient statistic, loss objective or gradient) is perturbed with noise that scales with the functional's *sensitivity*. The sensitivity $S(h)$ of a functional $h : \mathcal{D} \to \mathbb{R}$ is the maximum difference between the values of the function on any pair of neighboring data sets $D$ and $D'$, $S(h) = \max_{D,D'} \|h(D) - h(D')\|$. As the bound of statistical functionals given arbitrary data sets is typically unknown, their sensitivity is determined by assuming bounded input features [18] or parameter spaces [81]. In practise, noise is added either directly to the estimate $\hat{\theta}$ or the empirical loss function, skewing the interpretation of the released statistical estimates in ways that cannot be explained by probabilistic modelling assumptions. In differentially private stochastic gradient descent [DPSGD; 1], for example, the sensitivity of the mini-batch gradient in each update step is bounded by clipping the gradients of single batch observations before averaging. The averaged mini-batch gradient is then perturbed with noise that scales inversely with the gradients' clipping norm. The repercussions of so doing can be detrimental. The effects of statistical bias within DP estimation have been subject to recent scrutiny, and bias mitigation approaches have been proposed [31].

An alternative to the sensitivity method, is one-posterior sampling [OPS; 81]. Instead of artificially introducing noise that biases the learning procedure, OPS takes advantage of the inherent uncertainty provided by sampling from Bayesian posterior distributions [81, 66, 30, 85, 21, 22, 86]. Given prior $\pi(\theta)$ and likelihood $f(D; \theta)$, the *random* OPS estimate corresponds to a single sample from the Bayesian posterior $\pi(\theta|D) \propto f(D; \theta)\pi(\theta)$. If one accepts the Bayesian inference paradigm, then a probabilistic interpretation of the data generative distribution can be leveraged to sample interpretable DP parameter estimates. Additionally, Bayesian maximum-a-posteriori estimates are generally associated with regularised maximum likelihood estimates, and OPS has been shown to consistently learn the data generating parameter [81]. Therefore, OPS provides a compelling method for DP estimation independently of the analyst's perspective on the Bayesian paradigm. Current approaches to OPS, however, only provide DP under unrealistic conditions where the log-likelihood is bounded [81] or unbounded but Lipschitz and convex [66] limiting their implementation beyond logistic regression models – see Table 1.

In this paper, we extend the applicability of OPS to more general prediction tasks, making it a useful alternative to the sensitivity method. To do so, we combine the ability of OPS to produce consistent estimates with a robustified general Bayesian posterior aimed at minimising the $\beta$-divergence [9] between the model $f(D_i; \theta)$ and the data generating process. We henceforth refer to this privacy mechanism as $\beta$D-Bayes. While previous research has studied the benefits of the $\beta$-divergence for learning model parameters that are robust to outliers and model misspecification [32, 34, 45, 48, 46], we leverage its beneficial properties for DP parameter estimation: A feature of the $\beta$D-Bayes posterior is that it naturally provides a pseudo-log likelihood with bounded sensitivity for popular classification and regression models without having to modify the underlying model $f(\cdot; \theta)$. Further, such a bound is often independent of the predictive model, which allows, for instance, the privatisation of neural networks without an analysis of their sensitivity or perturbation of gradients.

**Contributions.** Our contributions can thus be summarised as follows:
- By combining OPS with the intrinsically bounded $\beta$-divergence, we propose $\beta$D-Bayes, a DP mechanism that applies to a general class of inference models.
- $\beta$D-Bayes bounds the sensitivity of a learning procedure without changing the model, getting rid of the need to assume bounded feature spaces or clipping statistical functionals. If the model is correctly specified, the data generating parameter can be learned consistently.

- $\beta$D-Bayes improves the efficiency of OPS leading to improved precision for DP logistic regression.
- The general applicability of $\beta$D-Bayes facilitates OPS estimation for complex models such as Bayesian neural network (BNN) classification and regression models.
- We provide extensive empirical evidence by reporting the performance of our model and four relevant baselines on ten different data sets, for two different tasks; additionally analysing their sensitivity in the number of samples and the available privacy budget.

## 2   Background and related work

Here, we focus on the OPS literature that analyses the inherent privacy of perfect sampling from a Bayesian posterior [21, 81, 27, 66, 30]. Such a DP mechanism is theoretically appealing as it does not violate generative modelling assumptions. Current approaches have, however, been limited in their application due to strong likelihood restrictions. We start by outlining the weaknesses of traditional DP methods and the state-of-the-art for OPS, before we introduce $\beta$D-Bayes OPS.

OPS is distinct from other work on DP Bayesian inference that can be categorised as either analysing the DP release of posterior samples using variational inference [40, 44, 71], or Monte Carlo procedures [81, 60, 37, 83, 29, 73, 85]. We show that these methods can be extended to approximately sample from $\beta$D-Bayes in Section 4.

**DP via the sensitivity method**   According to the exponential mechanism [63], sampling $\tilde{\theta}$ with probability proportional to $\exp(-\epsilon\ell(D,\tilde{\theta})/(2S(h))$ for a loss function $\ell : \mathcal{D} \times \Theta \to \mathbb{R}$ is $(\epsilon, 0)$-DP. A particularly widely used instance is the sensitivity method [26] which adds Laplace noise with scale calibrated by the sensitivity of the estimator. For example, consider empirical risk minimisation for a $p$ dimensional parameter $\theta \in \Theta \subseteq \mathbb{R}^p$:

$$\hat{\theta}(D) := \operatorname*{arg\,min}_{\theta \in \Theta} \frac{1}{n} \sum_{i=1}^{n} \ell(D_i, f(\cdot; \theta)) + \lambda R(\theta) \tag{1}$$

where $\ell(D_i, f(\cdot; \theta))$ is the loss function, $R(\theta)$ is a regulariser, and $\lambda > 0$ is the regularisation weight. Chaudhuri et al. [18] show that $\tilde{\theta} = \hat{\theta}(D) + z, \mathbb{R}^p \ni z = (z_1, \ldots, z_p) \overset{\text{iid}}{\sim} \mathcal{L}\left(0, \frac{2}{n\lambda\epsilon}\right)$, where $\mathcal{L}(\mu, s)$ is a Laplace distribution with density $f(z) = \frac{1}{2s}\exp\{-|z - \mu|/s\}$, is $(\epsilon, 0)$-DP provided $R(\cdot)$ is differentiable and 1-strongly convex, and $\ell(y_i, \cdot)$ is convex and differentiable with gradient $|\ell'(D_i, \cdot)| < 1$. The negative log-likelihood of logistic regression with a $L_2$ regulariser satisfies these conditions when the features are standardised between 0 and 1 – see Section A.4.1 for more details. Relaxing the conditions of convexity, DPSGD [1], which adds calibrated noise to gradient evaluations within stochastic gradient descent, has arisen as a general purpose tool for empirical risk minimisation [62, 20]. To achieve bounded sensitivity, DPSGD clips each single gradient within the update step. Instead of bounding the sensitivity artificially by assuming bounded feature spaces or clipping data functions, Dwork and Lei [25] identified the promise of robust methods for DP-estimation. Robust estimation procedures [e.g 41, 35] provide parameter estimates that are less sensitive to any one observation and therefore the scale of the noise that needs to be added to privatise such an estimate is reduced. While the connection between robust statistical estimators and DP has thus been subject to extensive research [5, 6, 25, 77, 22, 55, 17, 57, 58], we are the first to consider it within Bayesian sampling to produce a generally applicable method for consistent estimation of model parameters.

**DP via Gibbs one-posterior sampling**   Sampling from a Bayesian posterior constitutes a particular case of the exponential mechanism [81, 21, 22, 86], leading to the proposal of OPS through Gibbs posterior sampling. If the log-likelihood $\log f(D; \theta)$ is such that $\sup_{D,\theta} |\log f(D; \theta)| \leq B$, then one sample from the Gibbs posterior

$$\pi^{w \log f}(\theta|D) \propto \pi(\theta) \exp\left\{ w \sum_{i=1}^{n} \log f(D_i; \theta) \right\} \tag{2}$$

with $w = \frac{\epsilon}{4B}$ is $(\epsilon, 0)$-DP [81]. Note $w = 1$ recovers the standard posterior and $w \neq 1$ provides flexibility to adapt the posterior to the level of privacy required. However, standard models for inference do not usually have bounded log-likelihoods. Even discrete models with bounded support such as logistic regression do not. To overcome this, Wang et al. [81] assume a bounded parameter space and accordingly 'scale-down' the data. However, parameter spaces are typically unbounded, and data scaling has to be done carefully to avoid information leakage or loss inefficiency. Nevertheless,

Table 1: Requirements of different DP estimation techniques; sorted in decreasing strength of the restrictions imposed on the likelihood.

| | Unbounded Features | Likelihood Restriction | Prior Restriction | $\delta$ | Unbiased+ Consistent |
|---|---|---|---|---|---|
| Foulds et al. [27] | ✗ | exponential family + bounded sufficient statistics | conjugate | 0 | ✓ |
| Bernstein and Sheldon [12, 13] | ✗ | exponential family + bounded sufficient statistics | conjugate | 0 | ✓ |
| Minami et al. [66] | ✗ | convex + Lipschitz log-density | strongly convex | $> 0$ | ✓ |
| Wang et al. [81] | ✗ | bounded log-density | proper | 0 | ✓ |
| Chaudhuri et al. [18] | ✗ | convex log-density | strongly convex | 0 | ✓ |
| Abadi et al. [1] | ✓ | bounded gradients | none | $> 0$ | ✗ |
| $\beta$D-Bayes | ✓ | bounded density | proper | 0 | ✓ |

their algorithm is shown to outperform the perturbation approach proposed by [18]. Some works [21, 22, 86] overcome the assumption of a bounded log-likelihood by relaxing the definition of DP allowing distant observations according to some metric to be distinguishable with greater probability. Such a relaxation, however, no longer guarantees individual privacy. Geumlek et al. [30] consider versions of the Gibbs posterior for exponential family models and generalised linear models but in the context of Rényi-DP (RDP) [68], a relaxation of DP that bounds the Rényi divergence with parameter $\alpha$ between the posterior when changing one observation. Minami et al. [66] generalise the result of Wang et al. [81] to show that one sample from (2) with $w = \frac{\epsilon}{2L}\sqrt{m_\pi/(1 + 2\log(1/\delta))}$ for convex $L$-Lipschitz log-likelihoods with $m_\pi$-strongly convex regulariser is $(\epsilon, \delta)$-DP. For logistic regression, we have $L = 2\sqrt{d}$ if the features are bounded between 0 and 1. This, however, requires a relaxation to $\delta > 0$.

**Sufficient statistics perturbation**  Foulds et al. [27] identified that OPS mechanisms based on the Gibbs posterior are data inefficient in terms of asymptotic relative efficiency. For exponential family models, Foulds et al. [27] and Zheng [87] propose a Laplace mechanism that perturbs the data's sufficient statistics and considers conjugate posterior updating according to the perturbed sufficient statistics. The privatisation of the statistics allows for the whole posterior to be released rather than just one sample, and the perturbation of the statistics is independent of $n$ allowing the amount of noise to become smaller relative to the sufficient statistics as $n \to \infty$ providing consistent inference. They compare directly to [81] showing improved inference for exponential family models. However, these results require bounded sufficient statistics. Extensions include [12, 13, 67, 40, 71, 51]. These methods are further limited to exponential families which do not include popular machine learning models e.g. logistic regression or neural networks. In this paper, we set out to generalise the applicability of OPS and also address its efficiency issues by moving away from the Gibbs posterior, making use of tools used within generalised Bayesian updating.

## 3 $\beta$D-Bayes one-posterior sampling

**Generalised Bayesian Updating and the beta-Divergence**  OPS has struggled as a general purpose tool for DP estimation as bounding the sensitivity of $\log f(x; \theta)$ is difficult. Motivated by model misspecification, Bissiri et al. [15] showed that the posterior update

$$\pi^\ell(\theta|D) \propto \pi(\theta) \exp\left\{-\sum_{i=1}^{n} \ell(D_i, f(\cdot; \theta))\right\}, \quad (3)$$

assigning high posterior density to parameters that achieved small loss on the data, provides a coherent means to update prior beliefs about parameter $\theta_0^\ell := \arg\min_{\theta \in \Theta} \int \ell(D, f(\cdot; \theta))g(D)dz$ after observing data $D \sim g(\cdot)$. The Gibbs posterior in (2) is recovered using the weighted negative log-likelihood $\ell(D, f(\cdot; \theta)) = -w \log f(D; \theta)$, and the standard posterior for $w = 1$. This demonstrates that Bayesian inference learns about $\theta_0^{\log f} := \arg\min_{\theta \in \Theta} \int \log f(D; \theta)g(D)dD = \arg\min_{\theta \in \Theta} \text{KLD}(g||f(\cdot; \theta))$ [11, 80].

The framework of [15] provides the flexibility to choose a loss function with bounded sensitivity. An alternative, loss function that continues to depend on $\theta$ through the likelihood $f(\cdot; \theta)$ is the

$\beta$-divergence loss [9] for $\beta > 1$

$$\ell^{(\beta)}(D, f(\cdot; \theta)) = -\frac{1}{\beta - 1} f(D; \theta)^{\beta - 1} + \frac{1}{\beta} \int f(\overline{D}; \theta)^{\beta} d\overline{D}, \qquad (4)$$

so called as $\arg\min_\theta \mathbb{E}_{D \sim g} \left[ \ell^{(\beta)}(D, f(\cdot; \theta)) \right] = \arg\min_\theta D_B^{(\beta)}(g||f(\cdot; \theta))$ where $D_B^{(\beta)}(g||f)$ is the $\beta$-divergence defined in Section A. The first term in (4) contains the negative likelihood, therefore parameters that make the data likely achieve low loss. However, it is raised to the power $\beta - 1 > 1$ prescribing relatively smaller loss to observations unlikely under that parameter than the log-likelihood. A key feature of (4) is that while $\lim_{f \to 0} -\log f = \infty$, $\lim_{f \to 0} -\frac{f^{\beta-1}}{\beta-1} = 0$ for $\beta > 1$. The second integral term only depends on the parameters and ensures the $\beta$D loss can learn the data generating parameters. Setting $\beta = 1$ recovers the negative log-likelihood. We refer to updating using (3) and loss (4) as $\beta$D-Bayes. Note that this update is general, and can be applied for any choice of prior $\pi(\theta)$, and density/mass function $f(y; \theta)$. Using (4) for inference was first proposed by Basu et al. [9] and extended by Ghosh and Basu [32] to the Bayesian paradigm. Because of its favourable robustness properties, the $\beta$-divergence has since been deployed for a variety of examples within modern statistical inference [e.g. 47, 34, 48, 33, 79] and deep learning [e.g. 3, 36, 2, 19, 48].

A particularly convenient feature of inference based on divergences is that they are uniquely minimised to 0 when $f = g$. Therefore, if there exists $\theta_0$ such that $g(\cdot) = f(\cdot; \theta_0)$, i.e. the model is correctly specified for $g$, then $\arg\min_{\theta \in \Theta} D_B^{(\beta)}(g(\cdot)||f(\cdot; \theta)) = \arg\min_{\theta \in \Theta} \text{KLD}(g(\cdot)||f(\cdot; \theta)) = \theta_0$, and the $\beta$D-Bayes posterior will learn about the same parameter as the Gibbs posterior (2). Further, the general Bernstein-von-Mises theorem for generalised posteriors [Theorem 4; 65] can be applied to the $\beta$D-Bayes posterior (see Theorem 3) to show that $\pi^{(\beta)}(\theta|D)$ is asymptotically Gaussian and concentrates around $\theta_0^{\ell^{(\beta)}} := \arg\min_{\theta \in \Theta} D_B^{(\beta)}(g(\cdot)||f(\cdot; \theta))$ as $n \to \infty$. This proves useful when establishing consistency and asymptotic efficiency, see Section A.2. When the model is misspecified, i.e. there exists no $\theta_0$ such that $g(\cdot) = f(\cdot; \theta_0)$, then $\theta_0^{\ell^{(\beta)}} \neq \arg\min_{\theta \in \Theta} \text{KLD}(g(\cdot)||f(\cdot; \theta))$ and the $\beta$D-Bayes posterior learns a different parameter to the standard posterior. However, several works have argued that it provides more desirable inference [32, 34, 47, 48] and decision-making [45] under model misspecification, and stability [46] to the specification of the model. We now show that we can leverage the favourable robustness properties of $\beta$D-Bayes to obtain general-use DP OPS estimates.

**DP one-posterior-sampling**    Rather than bounding $\log f(\cdot; \theta)$, we replace it in (3) with the $\beta$D loss from (4) which is naturally bounded when the density is bounded.

**Condition 1** (Boundedness of the model density/mass function). *The model density or mass function* $f(\cdot; \theta)$ *is such that there exists* $0 < M < \infty$ *such that* $f(\cdot; \theta) \leq M, \forall \theta \in \Theta$.

**Lemma 1** (Bounded sensitivity of the $\beta$D-Bayes loss). *Under Condition 1 the sensitivity of the* $\beta$D-*Bayes-loss for any* $\beta > 1$ *is* $\left| \ell^{(\beta)}(D, f(\cdot; \theta)) - \ell^{(\beta)}(D', f(\cdot; \theta)) \right| \leq \frac{M^{\beta-1}}{\beta-1}$.

Bounding $f$ rather than its logarithm is considerably more straightforward: discrete likelihoods are always bounded by 1, while continuous likelihoods can be guaranteed to be bounded under mild assumptions. We will see in Example 2 that such a bound can be provided in a Gaussian regression model by truncating the support of the variance parameter $\sigma^2$ from below. Under Condition 1, Theorem 1 proves $(\epsilon, 0)$-DP of $\beta$D-Bayes OPS. Theorem 2 establishes the consistency of $\beta$D-Bayes OPS and Proposition 1 establishes its efficiency, based on the definitions given by [81, 27]. Condition 3, stated fully in Section A.2 requires that the $\beta$D-Bayes loss can be approximated by a quadratic form and requires that as $n$ grows the $\beta$D-Bayes loss is uniquely minimied.

**Theorem 1** (Differential privacy of the $\beta$D-Bayes posterior). *Under Condition 1, a draw* $\tilde{\theta}$ *from the* $\beta$D-*Bayes posterior* $\pi^{\ell^{(\beta)}}(\theta|D)$ *in* (3) *is* $\left( \frac{2M^{\beta-1}}{\beta-1}, 0 \right)$-*differentially private.*

**Theorem 2** (Consistency of $\beta$D-Bayes sampling). *Under Condition 3, stated in Section A.2,*

1. *a posterior sample* $\tilde{\theta} \sim \pi^{\ell^{(\beta)}}(\theta|D)$ *is a consistent estimator of* $\theta_0^{\ell^{(\beta)}}$.

2. *if data* $D_1, \ldots, D_n \sim g(\cdot)$ *were generated such that there exists* $\theta_0$ *with* $g(D) = f(D; \theta_0)$, *then* $\tilde{\theta} \sim \pi^{\ell^{(\beta)}}(\theta|D)$ *for all* $1 < \beta < \infty$ *is consistent for* $\theta_0$.

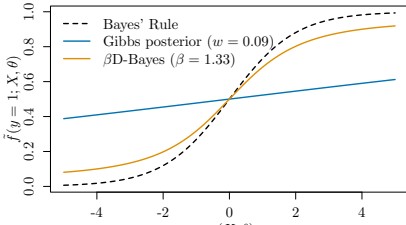 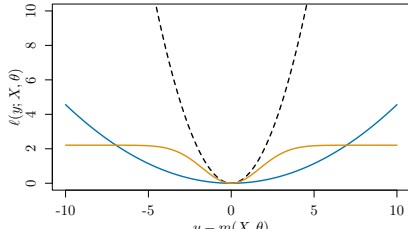

Figure 1: **Left:** Comparison of $\tilde{f}(y = 1; X, \theta) \propto \exp\{\ell(y = 1; X, \theta)\}$ for the Gibbs and $\beta$D-Bayes posteriors achieving DP with $\epsilon = 6$ with the standard logistic function for *binary classification* with $m(X, \theta) = X\theta$. The $\tilde{f}$ associated with the $\beta$D-Bayes posterior is closer to that of the Bayes posterior for any $m(X, \theta)$ than that of the Gibbs posterior for the same privacy level. **Right:** Comparison of $\ell(y; X, \theta)$ for the Gibbs posterior and the $\beta$D-Bayes posterior for *Gaussian regression* with the standard log-likelihood. The $\beta$D-Bayes posterior allows for DP estimation while the Gibbs posterior does not as any choice for $w > 0$ fails to bound the loss sensitivity.

**Proposition 1** (Asymptotic efficiency). *Under Condition 3, stated in Section A.2, $\tilde{\theta} \sim \pi^{\ell^{(\beta)}}(\theta | D)$ is asymptotically distributed as $\sqrt{n}(\tilde{\theta} - \theta_0^{\ell^{(\beta)}}) \overset{weakly}{\longrightarrow} \mathcal{N}(0, (H_0^{\ell^{(\beta)}})^{-1} K_0^{\ell^{(\beta)}} (H_0^{(\beta)})^{-1} + (H_0^{\ell^{(\beta)}})^{-1})$, where $K_0^{\ell^{(\beta)}}$ and $H_0^{\ell^{(\beta)}}$ are the gradient cross-product and Hessian matrices for the $\beta$D loss and are defined in (5) and (6).*

**OPS for classification and regression**    Note, unlike other methods, this DP guarantee does not require bounding features or response. We consider two explicit examples.

**Example 1** (Binary classification). *Consider a classifier for $y \in \{0, 1\}$ taking the logistic value of arbitrary function $m : \mathcal{X} \times \Theta \mapsto \mathbb{R}$, $p(y = 1; X, \theta) = 1/(1 + \exp(-m(X, \theta)))$, depending on predictors $X \in \mathcal{X}$ and parameters $\theta \in \Theta$. Clearly, $0 \leq f(y = 1; X, \theta) \leq 1$ independently of the functional form of $m(X, \theta)$, which guarantees $(\frac{2}{(\beta-1)}, 0)$-DP of the $\beta$D-Bayes OPS.*

Taking $m(X, \theta) = X^T \theta$ recovers logistic regression. Both the output perturbation of [18] and the Gibbs posterior (2) taking the Gibbs weight of [66] can also provide DP estimation for logistic regression. We show in Section 5 that $\beta$D-Bayes provides superior inference to these methods for the same privacy budget. Figure 1 (left) compares the standard Bayesian posterior for logistic regression with the Gibbs posterior ($w = 0.09$) and the $\beta$D-Bayes posterior ($\beta = 1.33$); both methods achieve DP estimation with $\epsilon = 6$. The $y$-axis is $\tilde{f}(y = 1; X, \theta) \propto \exp\{\ell(y = 1; X, \theta)\}$, the term that multiplies the prior in the updates (2) and (3). The figure shows that for the same privacy, the $\beta$D-Bayes update more closely resembles the standard Bayesian update allowing $\beta$D-Bayes OPS to produce more precise inference. Unlike [18] and [66], $\beta$D-Bayes also does not require bounding the features to guarantee DP.

Modern machine learning, however, often requires non-linear models, and $\beta$D-Bayes allows DP estimation for these as well. For example, we can choose $m(X, \theta) = NN(X, \theta)$ where NN is a neural network parameterised by $\theta$. See [70] for more details on Bayesian inference for neural networks. Unlike logistic regression, the log-likelihood of a neural network classifier is not convex and therefore the methods of [18] and [66] cannot be applied. This necessitates the application of DPSGD [1] which adds noise to minibatch gradient evaluations in SGD and clips the gradients at some value to artificially bound their sensitivity–see [84] for a Bayesian extension. In contrast, $\beta$D-Bayes gets rid of the need to bound the neural network's sensitivity. It further allows for DP estimation beyond classification.

**Example 2** (Gaussian regression). *Consider a Gaussian model $f(y; X, \theta, \sigma^2) = \mathcal{N}(y; m(X, \theta), \sigma^2)$ regressing univariate $y \in \mathbb{R}$ on predictors $X \in \mathcal{X}$ using any mean function $m : \mathcal{X} \times \Theta \to \mathbb{R}$ and parameter $\theta \in \Theta$ where $\sigma^2 > 0$ is the residual variance. Provided there exists a lower bound $0 < s^2 < \sigma^2$, then $0 \leq f(y; X, \theta, \sigma^2) \leq 1/(\sqrt{2\pi}s)$ independent of $m(X, \theta)$. $\beta$D-Bayes OPS for such a model is $(\epsilon, 0)$-DP with $\epsilon = 2/((\beta - 1)(\sqrt{2\pi}s)^{(\beta-1)})$.*

Ensuring Condition 1 for a Gaussian likelihood model requires that the support of the variance parameter is bounded away from 0. Such a bound is not limiting. For example, the standard conjugate

inverse-gamma prior puts vanishing prior density towards 0, and in situations where a natural lower bound is not available, adding independent and identically distributed zero-mean Gaussian noise with variance $s^2$ to the observed responses $y$ ensures this bound without changing the mean estimation.

One popular choice for the mean function is $m(X, \theta) = X^T\theta$ corresponding to standard linear regression. This is an exponential family model with conjugate prior, and DP estimation can be done using e.g. [13, 12]. They, however, require artificial bounds on the feature and response spaces. Again, the $\beta$D-Bayes estimation also holds for more complex mean functions such as neural networks – see [48] for $\beta$D-Bayes neural network regression. Figure 1 (right) illustrates how $\beta$D-Bayes bounds the sensitivity for Gaussian regression models while the Gibbs posterior cannot. The log-likelihood of the Gaussian distribution is unbounded and while multiplying this with $w < 1$ reduces the slope, this does not change its bound. The $\beta$D-Bayes, on the other hand, provides a bounded loss function.

## 4 Extending the privacy beyond one-posterior sampling

While OPS has been shown to provide DP guarantees for perfect samples, the OPS posterior is typically not available in closed form and as a result, some approximate sampling methods such as Markov Chain Monte Carlo (MCMC) are required. Note that this is not only a limitation of OPS, but sampling from the intractable exponential mechanism [63, 75] in general. Proposition 2 taken from [66] investigates the DP properties of the approximation.

**Proposition 2** (Proposition 12 of [66])**.** *If sampling from $\pi(\theta|D)$ is $(\epsilon, \delta)$-DP and for all $D$ there exists approximate sampling procedure $p_D(\theta)$ with $\int |\pi(\theta|D) - p_D(\theta)|d\theta \leq \gamma$, then sampling from $p_D(\theta)$ is $(\epsilon, \delta')$-DP with $\delta' = \delta + (1 + e^\epsilon)\gamma$.*

Proposition 2 establishes that if the MCMC chain has converged to the target distribution i.e. $\gamma \approx 0$, then the DP of the exact posterior is shared by the approximate sampling. Proposition 3 of [81] is the same result for $\delta = 0$. For the sufficiently well-behaved Gibbs posteriors (i.e. with Lipschitz convex loss function and strongly convex regulariser), Minami et al. [66] provide an analytic stepsize and number of iterations, $N$, that guarantees an (unadjusted) Langevin Monte-Carlo Markov Chain is within $\gamma$ of the target. While the Gibbs posterior for logistic regression satisfies these conditions, the Gibbs posterior for more general tasks and the $\beta$D-Bayes loss will in general not be convex.

Previous contributions [81, 66, 29] have assumed that the MCMC kernel has converged. Seeman et al. [75] observed that if the MCMC algorithm is geometrically ergodic achieving a $\delta'$ smaller than order $1/n$ and preventing data leakage requires the chain to be run for at least order $N = \log(n)$ iterations. For our experiments we used the No-U-turn Sampler [NUTS; 38] version of Hamiltonian Monte Carlo [HMC; 24] implemented in the `stan` probabilistic programming language [16]. The geometric ergodicity of HMC was established in [59] and `stan` provides a series of warnings that indicate when the chain does not demonstrate geometric ergodicity [14]. Running `stan` for sufficiently many iterations to not receive any warnings provides reasonable confidence of a negligible $\delta'$. As an alternative to measuring convergence, we below review approximate DP sampling approaches where $\beta$D-Bayes can be applied to guarantee bounded density. We hope that the emergence of $\beta$D-Bayes, as a general purpose OPS method to provide consistent DP estimation, motivates further research into private sampling methods for OPS.

**DP MCMC methods** Alternatively to attempting to release one sample from the exact posterior, much work has focused on extending OPS to release a Markov chain that approximates the Gibbs posterior, incurring per iteration privacy costs. Examples include the privatisation of Stochastic Gradient Langevin Dynamics [SGLD; 82, 81], the penalty algorithm [83], Hamiltonian Monte Carlo [DP-HMC 73], Baker's acceptance test [37, 8, 76], and rejection sampling [7]. Similar to DPSGD, these algorithms all rely on either subsampled estimates of the model's log-likelihood or its gradient to introduce the required noise and run the chain until the privacy budget has been used up. Just like the Gibbs posterior (2), the $\beta$D-Bayes posterior–using (4) and (3)–also contains a sum of loss terms that can be estimated via subsampling, making the $\beta$D-Bayes similarly amenable. What is more, many of these algorithms previously listed [85, 83, 37, 73, 81, 56, 84] require the boundedness of the sensitivity of the log-likelihood, or its derivative. Proposition 3 shows that under Condition 1 which ensures that $\beta$D-Bayes OPS is DP, or for gradient based samplers Condition 2, many of these DP samplers can be used to sample from the $\beta$D-Bayes posterior without compromising DP.

**Condition 2** (Boundedness of the model density/mass function gradient). *The model density or mass function $f(\cdot; \theta)$ is such that there exists $0 < G^{(\beta)} < \infty$ such that $\left\| \nabla_\theta f(D; \theta) \times f(D; \theta)^{\beta-2} \right\|_\infty \leq G^{(\beta)}, \forall \theta \in \Theta$.*

**Proposition 3** (DP-MCMC methods for the $\beta$D-Bayes-Posterior). *Under Condition 1, the penalty algorithm [Algorithm 1; 83], DP-HMC [Algorithm 1; 73] and DP-Fast MH [Algorithm 2; 85] and under Condition 2 DP-SGLD [Algorithm 1; 56] can produce $(\epsilon, \delta)$-DP estimation from the $\beta$D-Bayes posterior with $\delta > 0$ without requiring clipping.*

**Perfect sampling**    An alternative approach is to seek to modify MCMC chains to allow for exact samples from the posterior. Seeman et al. [75] privatise the perfect sampling algorithm of [54], which introduces an 'artificial atom' $a$ into the support of the target and uses an augmented MCMC kernel to move between the atom and the rest of the support. While we believe these approaches to be very promising and in principle trivially applicable to $\beta$D-Bayes OPS, Lee et al. [54] find considerable instability to choices for the underlying MCMC chain, and therefore more investigation is required.

**Attacking one-posterior samples**    In order to quantify the data leakage from approximate sampling schemes, we run the strongest privacy attacks for DP auditing, namely membership inference attacks (MIA). In MIA, an adversary tries to predict whether an observation was part of the training data set or not. This attack corresponds directly to the DP guarantee presented in Definition 1: Given any two neighbouring data sets, $D$ and $D'$, an attacker should not be able to confidently predict which data set was used in training if they observe the final statistical estimate $\tilde{\theta}$. Indeed, Jagielski et al. [43] have shown that the false positive and false negative rates of MIA attacks can be used to audit the DP of an algorithm directly. In the pursuit of tight auditing of DP algorithms, Nasr et al. [69] have proposed worst-case attacks where $|D| + |D'| = 1$ are chosen to maximise attack performance. These attacks are beneficial to uncover whether an algorithm violates its promised DP guarantees as published works have repeatedly been shown to suffer from faulty implementations or mistakes in proving DP [69, 78]. We are the first to consider such attacks for OPS.

For a number of rounds, we 1) generate two neighbouring data sets, $D$ and $D'$, 2) sample $m \sim$ Bernoulli$(0.5)$, 3) if $m = 1$ return $\tilde{\theta}(D')$ or if $m = 0$ return $\tilde{\theta}(D)$, and then 4) predict given Remark 1 which data set $\tilde{\theta}$ was trained on. Without loss of generality, we assume $m = 1$. We follow [74] in defining the objective of the MIA attack as $\mathcal{M}(\tilde{\theta}, D, D') := p(m = 1; \tilde{\theta}, D, D')$, i.e. the probability that $\tilde{\theta}$ was trained on $D'$ after observing $\tilde{\theta}$, $D$, and $D'$.

**Remark 1.** *Let $p(\tilde{\theta}|D)$ be the density of the privacy mechanism—i.e the Laplace density for [18] or the posterior (i.e. (2) or (3)) for OPS. An attacker estimating $\mathcal{M}(\tilde{\theta}, D, D') = \frac{p(\tilde{\theta}|D')}{(p(\tilde{\theta}|D) + p(\tilde{\theta}|D'))}$ is Bayes optimal. For OPS, $\mathcal{M}(\tilde{\theta}, D, D') = \exp\{\ell(D_l'; f(\cdot; \tilde{\theta})) - \ell(D_l; \tilde{\theta})\} \int \exp\{\ell(D_l; f(\cdot; \theta)) - \ell(D_l'; f(\cdot; \theta))\}\pi(\theta|D)d\theta$ where $D, D'$ s.t. $D \setminus D' = \{D_l\}$ and $D' \setminus D = \{D_l'\}$ (see Appendix A.5).*

## 5    Experimental results

Appendix B contains additional experimental details and results. Our code can be found at https://github.com/sghalebikesabi/beta-bayes-ops.

**Data sets**    The evaluations are conducted on simulated and UCI [23] data sets. For the former, we generate $d$-dimensional features from a multivariate normal distribution with the identity matrix as covariance matrix, sample the model parameters from a normal distribution with mean 0 and standard deviation 3 (i.e. a $d$ dimensional vector for the logistic regression), and simulate the label according to the assumed likelihood model. For the latter, we have included the two classification data sets that were previously analysed in other applications of OPS (`adult` and `abalone`) [66, 81], in addition to other popular UCI data sets. We report the test performance on random test splits that constitute 10% of the original data. We further scale the features in all data sets to lie within 0 and 1. This is a requirement for the methods proposed by [81, 66, 18], whereas $\beta$D-Bayes guarantees DP for unbounded data.

**Logistic regression**    Figure 2 compares $\beta$D-Bayes, Chaudhuri et al. [18] and Minami et al. [66] (with $\delta = 10^{-5}$). Note that [18] still presents a widely-used implementation of DP logistic regression [39]. We consider two implementations of [18]: one where $\lambda = 1/(9n)$ in (1) decreases in the

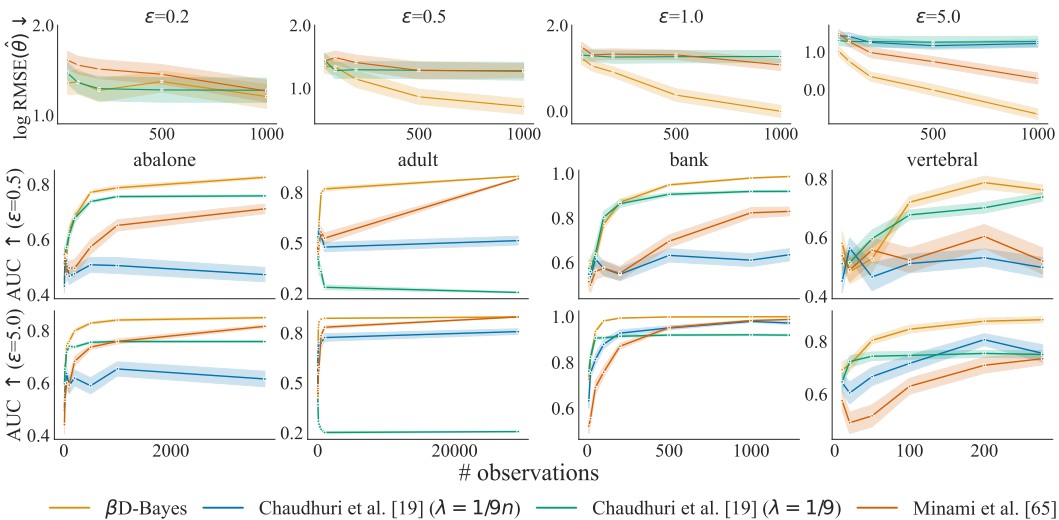

Figure 2: Parameter log RMSE and test-set ROC-AUC of DP estimation for logistic regression as the number of observations $n$ increases. We have upper bounded the axis of the log RMSE $(\hat{\theta})$, as [18] performs poorly when the regularisation term decreases in $n$.

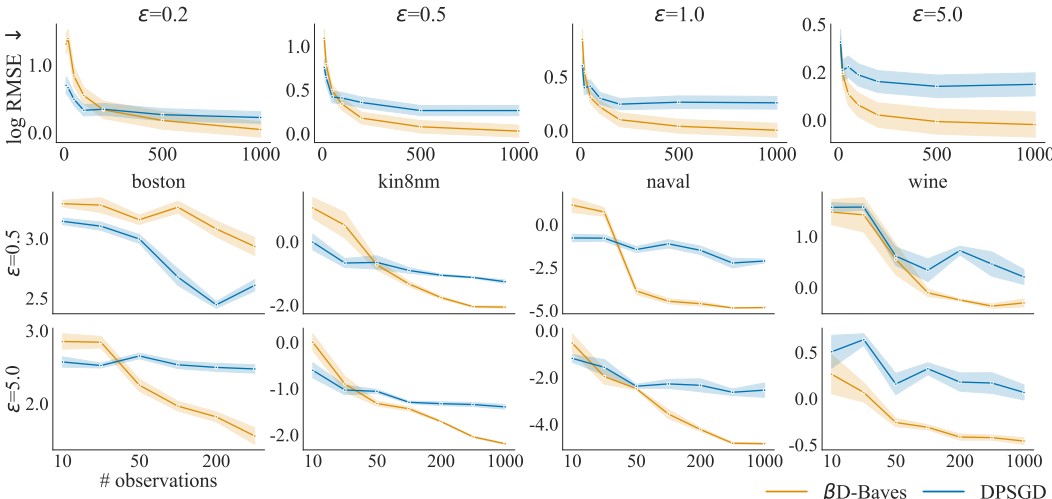

Figure 3: Test set predictive log RMSE of DP estimation for neural network regression as the number of observations $n$ increases on simulated and UCI data. See Appendix Figure 6 for a comparison of each method with its private counterpart.

number of samples, consistent with the Bayesian paradigm the effect of the regulariser diminishes as $n \to \infty$ and corresponds to a Gaussian prior with variance of 3, and another where $\lambda = 1/9$ is fixed - see A.4.1 for further discussion. The first leads to unbiased but inconsistent DP-inference while the second is consistent at the cost of also being biased. We report the RMSE between the estimated parameter $\tilde{\theta}(D)$ and the true data generating parameter on the simulated data sets, and the ROC-AUC on UCI data. For a compairson of each method with its non-private equivalent, please see Figure 6 in the appendix. We also introduce a new metric, termed *correct sign accuracy* that computes the proportion of coefficients that are 'correctly' estimated. Please refer to Figures 7 and 8 in the appendix.

In simulations, we observe that as $n$ increases, $\beta$D-Bayes achieves the smallest RMSE, illustrating the increased efficiency we argued for in Section 3. The extent to which $\beta$D-Bayes dominates is greater for large $\epsilon$. The ROC-AUC curves show that this better estimation of parameters generally corresponds to greater ROC-AUC. For the UCI data we see that for increasing $n$ $\beta$D-Bayes outperforms the other methods and Figure 6 shows it achieves performance that is very close to the unprivatised analogue.

**Neural network regression** Figure 3 compares $\beta$D-Bayes and DPSGD [1] (with $\delta = 10^{-5}$) for training a one-layer neural network with 10 hidden units. The learning rate, number of iterations, and clipping norm for DPSGD were chosen using validation splits to maximise its performance. For small $n$ the DPSGD is preferable in both simulations and the UCI data, but as the number of observations increases, the test set predictive RMSE of $\beta$D-Bayes outperforms that of DPSGD. Note that DPSGD is currently the best-performing optimiser for DP neural network training [20]. It privatises each gradient step, guaranteeing privacy for each parameter update, while $\beta$D-Bayes only guarantees DP for a perfect sample from the Bayesian posterior. DPSGD thus provides stronger privacy guarantees, and is more computationally efficient as MCMC scales poorly to high-dimensional feature spaces. We hyperparameter-tune the number of epochs, learning rate, and batch size of DP-SGD on a validation data set and use the same parameters on SGD for a fair comparison. Thus, the performance of SGD could be improved by a different choice of hyperparameters.

**Membership inference attacks** We implement the Bayes optimal attacker for the case of logistic regression. As the solution of the logistic regression is not defined when only observations from a single class are present, we choose $D = \{(1, 1), (0, 0)\}$ and $D' = \{(-1, 1), (0, 0)\}$ to achieve optimal attack results. Figure 4 compares the attack success rate with the log RMSE achieved on simulated data ($n = 1000, d = 2$). As $\epsilon$ increases, the attack success rate of all methods increases and their RMSE decreases, except [18] with $\lambda = 1/9$ whose RMSE does not decrease because of bias. Fixing the attack success rates, $\beta$D-Bayes generally achieves the lowest RMSE. Importantly, $\beta$D-Bayes is more efficient than [18] for the same attack success rate, where [18] has exact DP guarantees.

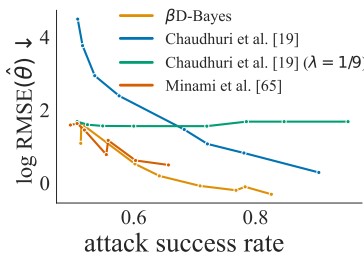

Figure 4: MIA attack success rates against log RMSE. Points correspond to values of $\epsilon$.

## 6 Discussion

We showed that $\beta$D-Bayes OPS produces consistent parameter estimates that are $(\epsilon, 0)$-DP provided that the model's density or mass function $f(\cdot; \theta)$ was bounded from above. $\beta$D-Bayes OPS improved the precision of DP inference for logistic regression and extends to more complex models, such as neural network classification and regression. Such extensions facilitate DP estimation for semi-parametric models where consistent inference for a linear predictor is required, but a complex model is used to capture the remaining variation [50]. Extensions of this work could consider dividing the privacy budget to release more than one sample paving the way for parameter inference as well as estimation. Bayesian inference with different divergences or discrepancies is becoming increasingly popular [28, 42, 4, 61], and their suitability for DP estimation could be investigated following our example and results. Further, the $\beta$D-loss is applicable beyond Bayesian methods, it could also be used in place of clipping gradients in algorithms such as DP-SGD [1].

The main limitation of OPS methods is that they prove DP for a sample from the exact posterior which is never tractable. Instead, MCMC samples approximate samples from the exact posterior and convergence of the MCMC sampler must be ensured to avoid leaking further privacy. Future work should investigate whether convergence guarantees are possible for $\beta$D-Bayes, taking advantage of its natural boundedness, and consider which of the DP-MCMC methods introduced in Section 4 makes best use of the privacy budget to sample a chain from $\beta$D-Bayes. A further limitation of OPS is the computational burden required to produce posterior samples, particularly in larger neural networks with many parameters. Such a cost and is mitigated by the fact that inference can only be run once to avoid leaking privacy and that only one posterior sample is required. But further research is required to tackle these computational challenges, including scaling MCMC to large neural networks or developing DP variational inference approaches [40, 44, 71] for the $\beta$D-Bayes posterior. We hope that the existence of a general-purpose method for DP estimates through Bayesian sampling with improved performance can stimulate further research in these directions.

## Acknowledgements

A thank you to Anna Menacher, Dan Moss and the anonymous reviewers for valuable comments on earlier drafts of the paper. JJ was funded by Juan de la Cierva Formación fellowship FJC2020-046348-I. SG was a PhD student of the EPSRC CDT in Modern Statistics and Statistical Machine Learning (EP/S023151/1), and also received funding from the Oxford Radcliffe Scholarship and

Novartis. CH was supported by The Alan Turing Institute, Health Data Research UK, the Medical Research Council UK, the EPSRC through the Bayes4Health programme Grant EP/R018561/1, and AI for Science and Government UK Research and Innovation (UKRI).

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

# Appendix A    Definitions, proofs, and related work

Here, we provide missing definitions of the Kullback-Leibler divergence (KLD) and the $\beta$-divergence ($\beta$D).

**Definition 2** (Kullback-Leibler divergence [52]). *The* KLD *between probability densities $g(\cdot)$ and $f(\cdot)$ is given by*

$$\text{KLD}(g||f) = \int g(x) \log \frac{g(x)}{f(x)} dx.$$

**Definition 3** ($\beta$-divergence [9, 64]). *The $\beta$D is defined as*

$$D_B^{(\beta)}(g||f) = \frac{1}{\beta(\beta-1)} \int g(x)^\beta dx + \frac{1}{\beta} \int f(x)^\beta dx - \frac{1}{\beta-1} \int g(x)f(x)^{\beta-1} dx,$$

*where $\beta \in \mathbb{R} \setminus \{0, 1\}$. When $\beta \to 1$, $D_B^{(\beta)}(g(x)||f(x)) \to \text{KLD}(g(x)||f(x))$.*

The $\beta$-divergence has often been referred to as the *density-power divergence* in the statistics literature [9] where it is also parameterised with $\beta_{DPD} = \beta - 1$.

Intuition for how $\beta$D-Bayes provides DP estimation is provided in Figure 5 which shows the influence of adding an observation $y$ at distance $|y - \mu|$ from the posterior mean $\mu$ when updating using a Gaussian distribution under KLD-Bayes and $\beta$D-Bayes. Here, influence is defined by [53] as the Fisher–Rao divergence between the posterior with or without that observation and can be easily estimated using an MCMC sample from the posterior without the observation. The influence of observations under KLD-Bayes is steadily increasing, making the posterior sensitive to extreme observations and therefore leaking their information. Under $\beta$D-Bayes, the influence initially increases before being maximised at a point depending on the value of $\beta$, before decreasing to 0. Therefore, each observation has bounded influence on the posterior, allowing for DP estimation.

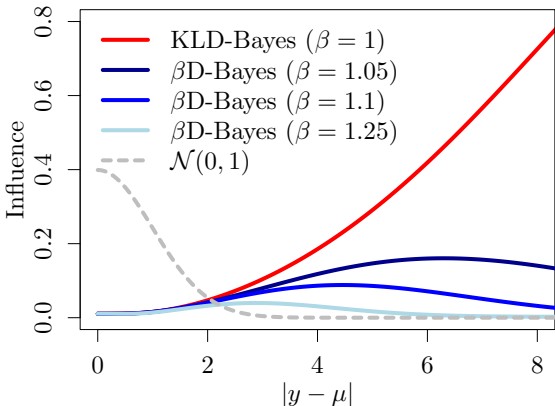

Figure 5: The influence of adding an observation $y$ at distance $|y - \mu|$ from the posterior mean $\mu$ on the posterior conditioned on a sample of 1000 points from a $\mathcal{N}(0, 1)$ when fitting a $\mathcal{N}(\mu, \sigma^2)$.

## A.1    Notation

Before proving the paper's results, we first define all notation used.

For likelihood $f(\cdot; \theta)$, prior $\pi(\theta)$ and data $D = \{D_i\}_{i=1}^n$ from data generating process $g$, the posterior, Gibbs posterior parameters by $w > 0$ and $\beta$D-Bayes posterior for $\beta > 1$ are given by

$$\pi(\theta|D) \propto \pi(\theta) \prod_{i=1}^n f(D_i; \theta)$$

$$\pi^{w \log f}(\theta|D) \propto \pi(\theta) \exp \left\{ w \sum_{i=1}^n f(D_i; \theta) \right\}$$

$$\pi^{\ell^{(\beta)}}(\theta|D) \propto \pi(\theta) \exp \left\{ -\sum_{i=1}^n \ell^{(\beta)}(D_i, f(\cdot; \theta)) \right\}$$

where

$$\ell^{(\beta)}(D, f(\cdot; \theta)) = -\frac{1}{\beta - 1} f(D; \theta)^{\beta-1} + \frac{1}{\beta} \int f(\overline{D}; \theta)^{\beta} d\overline{D}.$$

The *population* loss minimising parameters are

$$\theta_0^{\log f} := \operatorname*{arg\,min}_{\theta \in \Theta} \int \log f(D; \theta) g(D) dD = \operatorname*{arg\,min}_{\theta \in \Theta} \mathrm{KLD}(g || f(\cdot; \theta))$$

$$\theta_0^{\ell^{(\beta)}} := \operatorname*{arg\,min}_{\theta \in \Theta} \int \ell^{(\beta)}(D, f(\cdot; \theta)) g(D) dD = \operatorname*{arg\,min}_{\theta \in \Theta} D_B^{(\beta)}(g || f(\cdot; \theta))$$

and their sample estimates

$$\hat{\theta}_n^{\log f} := \operatorname*{arg\,min}_{\theta \in \Theta} \sum_{i=1}^{n} \log f(D_i; \theta)$$

$$\hat{\theta}_n^{\ell^{(\beta)}} := \operatorname*{arg\,min}_{\theta \in \Theta} \sum_{i=1}^{n} \ell^{(\beta)}(D_i, f(\cdot; \theta))$$

If the model is well specified for the data generating process $g$ then we write $g(\cdot) = f(\cdot; \theta_0)$ and $\theta_0 = \arg\min_{\theta \in \Theta} \mathrm{KLD}(g || f(\cdot; \theta)) = \arg\min_{\theta \in \Theta} D_B^{(\beta)}(g || f(\cdot; \theta))$.

Lastly, our asymptotic results require the definitions of the Hessian and gradient cross-product matrices

$$\hat{H}_n^{\ell^{(\beta)}} := \left( \frac{\partial}{\partial \theta_i \partial \theta_j} \frac{1}{n} \sum_{i=1}^{n} \ell^{(\beta)}(D_i, f(\cdot; \theta_0^{\ell^{(\beta)}})) \right)_{i,j}$$

$$H_0^{\ell^{(\beta)}} := \left( \frac{\partial}{\partial \theta_i \partial \theta_j} \mathbb{E}_D \left[ \ell^{(\beta)}(D, f(\cdot; \theta_0^{\ell^{(\beta)}})) \right] \right)_{i,j} \tag{5}$$

$$K_0^{\ell^{(\beta)}} := \left( \frac{\partial}{\partial \theta_i} \mathbb{E}_D \left[ \ell^{(\beta)}(D; f(\cdot; \theta_0^{\ell^{(\beta)}})) \right] \frac{\partial}{\partial \theta_j} \mathbb{E}_D \left[ \ell^{(\beta)}(D; f(\cdot; \theta_0^{\ell^{(\beta)}})) \right] \right)_{i,j}. \tag{6}$$

### A.2 Bernstein-von Mises theorem for $\beta$D-Bayes

The general Bernstein-von Mises theorem for generalised posteriors [Theorem 4; 65] can be applied to the $\beta$D-Bayes posterior. We first state the required Condition 3 before stating the result.

**Condition 3** (Assumptions of Theorem 4 of [65] for the $\beta$D-Bayes). *Fix $\theta_0^{\ell^{(\beta)}} \in \mathbb{R}^p$ and let prior $\pi(\theta)$ is continuous at $\theta_0^{\ell^{(\beta)}}$ with $\pi(\theta_0^{\ell^{(\beta)}}) > 0$. Let $f_n^{(\beta)} : \mathbb{R}^p \to \mathbb{R}$ with $f_n^{(\beta)}(\theta) = \frac{1}{n} \sum_{i=1}^{n} \ell^{(\beta)}(D_i, f(\cdot; \theta)$ for $n \in \mathbb{N}$ and assume:*

*(1) $f_n^{(\beta)}$ can be represented as*

$$f_n^{(\beta)}(\theta) = f_n^{(\beta)}(\hat{\theta}_n^{\ell^{(\beta)}}) + \frac{1}{2}(\theta - \hat{\theta}_n^{\ell^{(\beta)}})^T \hat{H}_n^{\ell^{(\beta)}}(\theta - \hat{\theta}_n^{\ell^{(\beta)}}) + r_n^{(\beta)}(\theta - \hat{\theta}_n^{\ell^{(\beta)}})$$

*where $\hat{\theta}_n^{\ell^{(\beta)}} \in \mathbb{R}^p$ and $\hat{\theta}_n^{\ell^{(\beta)}} \to \theta_0^{\ell^{(\beta)}}$, with $\hat{H}_n^{\ell^{(\beta)}} \to H_0^{\ell^{(\beta)}}$ for positive definite $H_0^{\ell^{(\beta)}}$, and $r_n^{(\beta)} : \mathbb{R}^p \to \mathbb{R}$ has the following property: there exist $\epsilon_0, c_0 > 0$ such that for all $n$ sufficiently large, for all $x \in B_{\epsilon_0}(0)$, we have $|r_n^{(\beta)}(x)| \le c_0 |x|^3$.*

*(2) For any $\epsilon > 0$, $\liminf_n \inf_{\theta \in B_\epsilon(\hat{\theta}_n^{\ell^{(\beta)}})^c} (f_n^{(\beta)}(\theta) - f_n^{(\beta)}(\hat{\theta}_n^{\ell^{(\beta)}})) > 0$, where $B_r(x_0) = \{x \in \mathbb{R}^D : |x - x_0| < r\}$*

Condition 3 (1) requires that the $\beta$D-Bayes loss can be approximated by a quadratic form and (2) requires that as $n$ grows the $\beta$D-Bayes loss is uniquely minimied at $\hat{\theta}_n^{\ell^{(\beta)}}$.

**Theorem 3** (Theorem 4 of [65] for the $\beta$D-Bayes). *Assume Condition 3. Define $\pi_n^{\ell^{(\beta)}} := \pi^{\ell^{(\beta)}}(\theta | D = \{D_1, \ldots, D_n\})$. Then*

*(i) The $\beta$D-Bayes posterior concentrates on $\theta_0^{(\beta)}$*

$$\int_{B_\varepsilon(\theta_0^{(\beta)})} \pi_n^{\ell^{(\beta)}}(\theta)d\theta \xrightarrow[n\to\infty]{} 1 \tag{7}$$

*where $B_r(x_0) = \{x \in \mathbb{R}^p : |x - x_0| < r\}$*

*(ii) The $\beta$D-Bayes posterior is asymptotically Gaussian*

$$\int \left| \tilde{\pi}_n^{\ell^{(\beta)}}(\phi) - \mathcal{N}_p\left(\phi; 0, (H_0^{\ell^{(\beta)}})^{-1}\right) \right| d\phi \xrightarrow[n\to\infty]{} 0 \tag{8}$$

*where $\tilde{\pi}_n^{\ell^{(\beta)}}$ denotes the density of $\sqrt{n}(\tilde{\theta} - \hat{\theta}_n^{\ell^{(\beta)}})$ when $\tilde{\theta} \sim \pi_n^{\ell^{(\beta)}}$, $\mathcal{N}_p(x; \mu, \Sigma)$ denotes the p-dimensional multivariate Gaussian distribution with mean vector $\mu$ and covariance matri $\Sigma$*

*Proof.* The result is proved as a direct application of Theorem 4 of [65] with $f_n(\theta) = \frac{1}{n}\sum_{i=1}^n \ell^{(\beta)}(D_i, f(\cdot; \theta))$ being the $\beta$D-Bayes loss function. $\square$

Theorem 3 shows that the $\beta$D-Bayes posterior concentrates on $\theta_0^{(\beta)}$ and converges to a Gaussian distribution centred around the $\beta$D minimising parameter $\theta_0^{\ell^{(\beta)}}$ in total variation distance.

## A.3 Proofs

### A.3.1 Proof of Lemma 1

**Lemma 1** (Bounded sensitivity of the $\beta$D-Bayes loss). *Under Condition 1 the sensitivity of the $\beta$D-Bayes-loss for any $\beta > 1$ is $\left| \ell^{(\beta)}(D, f(\cdot; \theta)) - \ell^{(\beta)}(D', f(\cdot; \theta)) \right| \leq \frac{M^{\beta-1}}{\beta-1}$.*

*Proof.* By (4), for $\beta > 1$

$$\left| \ell^{(\beta)}(D, f(\cdot; \theta)) - \ell^{(\beta)}(D', f(\cdot; \theta)) \right| = \frac{1}{\beta - 1}\left( f(D'; \theta)^{\beta-1} - f(D; \theta)^{\beta-1} \right)$$

$$\leq \max_D \frac{1}{\beta - 1} f(D; \theta)^{\beta-1}$$

$$\leq \frac{M^{\beta-1}}{\beta - 1}$$

$\square$

### A.3.2 Proof of Theorem 1

**Theorem 1** (Differential privacy of the $\beta$D-Bayes posterior). *Under Condition 1, a draw $\tilde{\theta}$ from the $\beta$D-Bayes posterior $\pi^{(\beta)}(\theta|D)$ in (3) is $\left( \frac{2M^{\beta-1}}{\beta-1}, 0 \right)$-differentially private.*

*Proof.* Define $D = \{D_1, \ldots, D_n\}$, $D' = \{D'_1, \ldots, D'_n\}$ and let $j$ be the index such that $D_j \neq D'_j$ with $D_i = D'_i$ for all $i \neq j$. Firstly, the normalising constant of the $\beta$D-Bayes posterior combining (3) with (4) is

$$P^{(\beta)}(D) := \int \pi(\theta) \exp\left\{ -\sum_{i=1}^n \ell^{(\beta)}(\theta, D_i) \right\} d\theta \tag{9}$$

Then,

$$\log \frac{\pi^{(\beta)}(\theta|D)}{\pi^{(\beta)}(\theta|D')} = \sum_{i=1}^n \ell^{(\beta)}(D'_i, f(\cdot; \theta)) - \sum_{i=1}^n \ell^{(\beta)}(D_i, f(\cdot; \theta)) + \log \frac{P^{(\beta)}(D')}{P^{(\beta)}(D)}$$

$$= \ell^{(\beta)}(D'_j; f(\cdot; \theta)) - \ell^{(\beta)}(D_j; f(\cdot; \theta)) + \log \frac{P^{(\beta)}(D')}{P^{(\beta)}(D)}$$

Now, by Condition 1 and Lemma 1,

$$\ell^{(\beta)}(D_j'; f(\cdot; \theta)) - \ell^{(\beta)}(D_j; f(\cdot; \theta)) \le \frac{M^{\beta-1}}{\beta-1},$$

and

$$P^{(\beta)}(D') = \int \exp\left\{-\sum_{i=1}^n \ell^{(\beta)}(D_i', f(\cdot; \theta))\right\} \pi(\theta)d\theta$$

$$= \int \exp\left\{\ell^{(\beta)}(D_j, f(\cdot; \theta)) - \ell^{(\beta)}(D_j', f(\cdot; \theta)) - \sum_{i=1}^n \ell^{(\beta)}(D_i, f(\cdot; \theta))\right\} \pi(\theta)d\theta$$

$$\le \exp\left\{\frac{M^{\beta-1}}{\beta-1}\right\} \int \exp\left\{-\sum_{i=1}^n \ell^{(\beta)}(D_i, f(\cdot; \theta))\right\} \pi(\theta)d\theta,$$

which combined provides that

$$\log \frac{\pi(\theta|D)}{\pi(\theta|D')} \le 2\frac{M^{\beta-1}}{\beta-1}.$$

$\square$

### A.3.3 Proof of Theorem 2

**Theorem 2** (Consistency of $\beta$D-Bayes sampling). *Under Conditions 3,*
1. *a posterior sample $\tilde\theta \sim \pi^{\ell^{(\beta)}}(\theta|D)$ is a consistent estimator of $\theta_0^{\ell^{(\beta)}}$.*
2. *if data $D_1, \ldots, D_n \sim g(\cdot)$ were generated such that there exists $\theta_0$ with $g(D) = f(D; \theta_0)$, then $\tilde\theta \sim \pi^{\ell^{(\beta)}}(\theta|D)$ for all $1 < \beta < \infty$ is consistent for $\theta_0$.*

*Proof.* For Part 1), define $B_r(x_0) = \{x \in \mathbb{R}^p : |x - x_0| < r\}$. Theorem 3 Part (i) proves that

$$\int_{B_\varepsilon(\theta_0^{(\beta)})} \pi_n^{\ell^{(\beta)}}(\theta)d\theta \xrightarrow[n\to\infty]{} 1 \tag{10}$$

for all $\varepsilon > 0$ and $\pi_n^{\ell^{(\beta)}} := \pi^{\ell^{(\beta)}}(\theta|D = \{D_1, \ldots, D_n\})$. This is enough to show that for $\tilde\theta \sim \pi^{\ell^{(\beta)}}(\theta|D) \to \theta_0^{\ell^{(\beta)}}$ in probability.

For Part 2), note that if $g(D) = f(D; \theta_0)$, then for all $1 < \beta < \infty$

$$\theta_0^{\ell^{(\beta)}} := \arg\min_{\theta\in\Theta} \mathbb{E}_g\left[\ell^{(\beta)}(D; f(\cdot; \theta))\right]$$

$$= \arg\min_{\theta\in\Theta} D_B^{(\beta)}(g||f(\cdot; \theta)) = \theta_0.$$

$\square$

### A.3.4 Proof of Proposition 1

**Proposition 1** (Asymptotic efficiency). *Under Conditions 3, $\tilde\theta \sim \pi^{\ell^{(\beta)}}(\theta|D)$ is asymptotically distributed as $\sqrt{n}(\tilde\theta - \theta_0^{\ell^{(\beta)}}) \xrightarrow{weakly} \mathcal{N}(0, (H_0^{\ell^{(\beta)}})^{-1}K_0^{\ell^{(\beta)}}(H_0^{\ell^{(\beta)}})^{-1} + (H_0^{\ell^{(\beta)}})^{-1})$, where $K_0^{\ell^{(\beta)}}$ and $H_0^{\ell^{(\beta)}}$ are the gradient cross-product and Hessian matrices for the $\beta$D loss and are defined in (5) and (6).*

*Proof.* Let $\tilde\theta \sim \pi^{\ell^{(\beta)}}(\theta|D)$. By Theorm 3 Part (ii),

$$\sqrt{n}(\tilde\theta - \hat\theta_n^{\ell^{(\beta)}}) \to \mathcal{N}(0, (H_0^{\ell^{(\beta)}})^{-1}).$$

By the asymptotic normality of $\hat\theta_n^{\ell^{(\beta)}}$ [10], we have that

$$\sqrt{n}(\hat\theta_n^{\ell^{(\beta)}} - \theta_0^{\ell^{(\beta)}}) \to^D \mathcal{N}(0, (H_0^{\ell^{(\beta)}})^{-1}K_0^{\ell^{(\beta)}}(H_0^{\ell^{(\beta)}})^{-1})$$

for

$$K_0^{\ell^{(\beta)}} := \left(\frac{\partial}{\partial\theta_i}\mathbb{E}_D\left[\ell^{(\beta)}(D; f(\cdot; \theta_0^{\ell^{(\beta)}}))\right] \frac{\partial}{\partial\theta_j}\mathbb{E}_D\left[\ell^{(\beta)}(D; f(\cdot; \theta_0^{\ell^{(\beta)}}))\right]\right)_{i,j}.$$

The result then comes from the asymptotic independence of $\tilde\theta - \hat\theta_n^{\ell^{(\beta)}}$ and $\hat\theta_n^{\ell^{(\beta)}}$ [see e.g. 81]. $\square$

### A.3.5 Proof of Proposition 3

Lemma 1 considers the sensitivity of the $\beta$D-Bayes loss to the change in one *observation*. DP-MCMC methods require the sensitivity to the change in the *parameter* which is provided by Lemma 2.

**Lemma 2** (Bounded parameter sensitivity of the $\beta$D-Bayes loss). *Under Condition 1 the parameter sensitivity of the $\beta$D-Bayes-loss for any $\beta > 1$ is $\left| \ell^{(\beta)}(D, f(\cdot; \theta)) - \ell^{(\beta)}(D, f(\cdot; \theta')) \right| \leq \frac{2\beta - 1}{\beta(\beta - 1)} M^{\beta - 1}$.*

*Proof.* By (4), for $\beta > 1$

$$\left| \ell^{(\beta)}(D, f(\cdot; \theta)) - \ell^{(\beta)}(D, f(\cdot; \theta')) \right|$$

$$= \left| -\frac{1}{\beta - 1} f(D; \theta)^{\beta - 1} + \frac{1}{\beta} \int f(\overline{D}; \theta)^{\beta} d\overline{D} + \frac{1}{\beta - 1} f(D; \theta')^{\beta - 1} - \frac{1}{\beta} \int f(\overline{D}; \theta')^{\beta} d\overline{D} \right|$$

$$\leq \max_{\theta, \theta'} \left\{ \frac{1}{\beta} \int f(\overline{D}; \theta)^{\beta} d\overline{D} + \frac{1}{\beta - 1} f(D; \theta')^{\beta - 1} \right\}$$

$$\leq \max_{\theta, \theta'} \left\{ \frac{1}{\beta} \mathbb{E}_{f(\overline{D}; \theta)} [f(\overline{D}; \theta)^{\beta - 1}] + \frac{1}{\beta - 1} M^{\beta - 1} \right\}$$

$$\leq \frac{1}{\beta} M^{\beta - 1} + \frac{1}{\beta - 1} M^{\beta - 1}$$

$$= \frac{2\beta - 1}{\beta(\beta - 1)} M^{\beta - 1}$$

$\square$

**Proposition 3** (DP-MCMC methods for the $\beta$D-Bayes-Posterior). *Under Condition 1, the penalty algorithm [Algorithm 1; 83], DP-HMC [Algorithm 1; 73] and DP-Fast MH [Algorithm 2; 85] and under Condition 2 DP-SGLD [Algorithm 1; 56] can produce $(\epsilon, \delta)$-DP estimation from the $\beta$D-Bayes posterior with $\delta > 0$ without requiring clipping.*

*Proof.* Algorithm 1 of [83] (Assumption A1), Algorithm 1 of [73] and Algorithm 2 of [85] (Assumptions 1 and 2) require a posterior whose log-likelihood has bounded *parameter* sensitivity. For $\beta$D-Bayes posterior, this requires the $\beta$D-Bayes-loss to have bounded *parameter* sensitivity which is provided by Condition 1 and Lemma 2.

Algorithm 1 of [56] requires a posterior whose log-likelihood has bounded gradient. For $\beta$D-Bayes posterior, this requires $\beta$D-Bayes-loss to have bounded gradient. Assuming we can interchange integration and differentiation, this requires

$$\left|\left| \nabla_{\theta} \ell^{(\beta)}(D; \theta) \right|\right|_{\infty} = \left|\left| \nabla_{\theta} f(D; \theta) \times f(D; \theta)^{\beta - 2} - \int \nabla_{\theta} f(\overline{D}; \theta) \times f(\overline{D}; \theta)^{\beta - 1} d\overline{D} \right|\right|_{\infty}$$

$$= \left|\left| \nabla_{\theta} f(D; \theta) \times f(D; \theta)^{\beta - 2} - \int \nabla_{\theta} f(\overline{D}; \theta) \times f(\overline{D}; \theta)^{\beta - 2} \times f(\overline{D}; \theta) d\overline{D} \right|\right|_{\infty}$$

$$\leq \max\{G^{(\beta)}, G^{(\beta)} M\},$$

as $\left|\left| \nabla_{\theta} f(D; \theta) \times f(D; \theta)^{\beta - 2} \right|\right|_{\infty} \leq G^{(\beta)}$ by Condition 2. $\square$

We show that Condition 2 is satisfied for logistic regression.

**Example 3** (Satisfying Condition 2 for logistic regression). *For Logistic regression introduced in Example 1*

$$f(y; \theta, X) = \frac{1}{(1 + \exp(-(2y - 1)X\theta))}$$

$$\nabla_{\theta} f(y; \theta, X) = \frac{(2y - 1)X}{(1 + \exp(-X\theta))^2}$$

$$f(y; \theta, X)^{\beta - 2} = (1 + \exp(-(2y - 1)X\theta))^{2 - \beta}$$

*and therefore,*

$$\left|\left|\nabla_\theta f(y;\theta,X) \times f(y;\theta,X)^{\beta-2}\right|\right|_\infty = \left|\left|\frac{(2y-1)X(1+\exp(-(2y-1)X\theta))^{2-\beta}}{(1+\exp(-(2y-1)X\theta))^2}\right|\right|_\infty$$

$$= \left|\left|\frac{(2y-1)X}{(1+\exp(-(2y-1)X\theta))^\beta}\right|\right|_\infty$$

$$\leq ||X||_\infty.$$

*Satisfying Condition 2 for logistic regression requires bounding of the features, a similar requirement to the* DP *methods of [81, 66, 18].*

## A.4   Related work

Here, we would like to extend our discussion of two important areas within the related work.

### A.4.1   Differentially private logistic regression

**Extensions to $L_1$-sensitivity**   Our presentation of the object perturbation of Chaudhuri et al. [18] in Section 2 differs slightly from its original presentation. Corollary 8 [18] assumes that $||\nabla_\theta \ell(D_i, f(\cdot;\theta))|| < 1$ which allows the bounding of the $L_2$-sensitivity of the empirical risk minimiser (1). Then adding *multivariate* Laplace noise $z' \sim \nu(z) \propto \exp(-\frac{n\lambda\epsilon}{2}||z'||_2)$ provides DP estimation. For logistic regression, $||\nabla_\theta \ell(D_i, f(\cdot;\theta))|| < 1$ requires that the features $X$ are such that $||X||_2 < 1$.

In order to provide a fairer comparison with [66], who require that $|\nabla_{\theta_j} \ell(D_i, f(\cdot;\theta)| < 1$, $j = 1, \ldots, p$, which for logistic regression requires $|X_j| < 1$ for $j = 1, \ldots, p$, we add *univariate* Laplace noise to each parameter estimate requiring us to bound the $L_1$ sensitivity of the empirical risk minimiser (1). This is a natural extension of the result of Chaudhuri et al. [18], but for completeness, we provide a proof that the approach presented in Section 2 is $(\epsilon, 0)$-DP.

**Theorem 4** (Modification of Output-Perturbation Chaudhuri et al. [18])**.** *Consider empirical risk minimisation for a $p$ dimensional parameter $\theta \in \Theta \subseteq \mathbb{R}^p$:*

$$\hat{\theta}(D) := \arg\min_{\theta \in \Theta} \frac{1}{n} \sum_{i=1}^{n} \ell(D_i, f(\cdot;\theta)) + \lambda R(\theta)$$

*where $\ell(D_i, f(\cdot;\theta))$ is convex and differentiable loss function with $|\nabla_{\theta_j} \ell(D_i, f(\cdot;\theta)| < 1$, $j = 1, \ldots, p$, $R(\theta)$ is a differentiable and 1-strongly convex regulariser, and $\lambda > 0$ is the regularisation weight. Then releasing*

$$\tilde{\theta} = \hat{\theta}(D) + z, \mathbb{R}^p \in z = (z_1, \ldots, z_p) \stackrel{iid}{\sim} \mathcal{L}\left(0, \frac{2}{n\lambda\epsilon}\right),$$

*where $\mathcal{L}(\mu, s)$ is a Laplace distribution with density $f(z) = \frac{1}{2s}\exp\{-|z-\mu|/s\}$, is $(\epsilon, 0)$-DP.*

*Proof.* Fix datasets $D_1$ and $D_2 = \{D_1 \setminus D_k\} \cup D_l$. Then define

$$\theta_k := \hat{\theta}(D_k) = \arg\min_{\theta \in \Theta} G_k(\theta)$$

$$G_k(\theta) := \frac{1}{n} \sum_{D \in D_k} \ell(D, f(\cdot;\theta) + \lambda R(\theta)$$

$$g(\theta) := G_1(\theta) - G_2(\theta)$$

$$= \frac{1}{n}\ell(D_k, f(\cdot;\theta) - \frac{1}{n}\ell(D_l, f(\cdot;\theta)$$

Lemma 7 of Chaudhuri et al. [18] then proves that

$$||\theta_1 - \theta_2||_2 \leq \frac{1}{\lambda} \max_{\theta \in \Theta} ||\nabla_\theta g(\theta)||_2$$

Now, as $|\nabla_{\theta_j}\ell(D_i, f(\cdot;\theta))| < 1, j = 1, \ldots, p$

$$\begin{aligned}
||\nabla_\theta g(\theta)||_2 &= \frac{1}{n}||\nabla_\theta\ell(D_k, f(\cdot;\theta) - \nabla_\theta\ell(D_l, f(\cdot;\theta))||_2 \\
&\leq \frac{2}{n}\max_D ||\nabla_\theta\ell(D, f(\cdot;\theta))||_2 \\
&= \frac{2}{n}\max_D \sqrt{\sum_{j=1}^p \nabla_{\theta_j}\ell(D, f(\cdot;\theta))} \\
&\leq \frac{2\sqrt{p}}{n}
\end{aligned}$$

The original result of Chaudhuri et al. [18] assumed that $||\nabla_\theta\ell(D_i, f(\cdot;\theta))||_2 < 1$ and they obtained $||\nabla_\theta g(\theta)||_2 < \frac{2}{n}$. Instead, we require a weaker element wise bound that $|\nabla_{\theta_j}\ell(D_i, f(\cdot;\theta))| < 1$ and note that

$$||\theta_1 - \theta_2||_2 \leq \frac{2\sqrt{p}}{\lambda} \Rightarrow ||\theta_1 - \theta_2||_1 \leq \frac{2}{n\lambda}.$$

Therefore by Proposition 1 of [26] we have that $\tilde{\theta} = \hat{\theta}(D) + (z_1, \ldots, z_p)$ with $z_j \sim \mathcal{L}\left(0, \frac{2}{n\epsilon\lambda}\right)$ is $(\epsilon, 0)$-DP. $\qquad\square$

Note, 1-strong convexity of an $L_2$ type regulariser requires that $R(\theta) = \frac{1}{2}||\theta||_2^2 = \frac{1}{2}\sum_{j=1}^p \theta_j^2$. This parametrisation is the default implementation in `sklearn`.

**Consistency**  Chaudhuri et al. [18] propose a regularised DP logistic regression, solving (1). The optimisation problem in (1) adds the regulariser to the average loss and as a result, the impact of the regulariser does not diminish as $n \to \infty$. Even though the scale of the Laplace noise decreases as $n$ grows, Chaudhuri et al. [18] consistently estimate a parameter that is not the data generating parameter. Alternatively, one could choose a regulariser $\lambda' := \frac{\lambda}{n}$ whose influence decreases as $n$ grows. This would allow for unbiased inference as $n \to \infty$ (assuming a Bayesian model with corresponding prior distribution), but the $n$ cancels in the scale of the Laplace noise and therefore the perturbation scale does not decrease in $n$, and the estimator is inconsistent. Choosing instead $\lambda' := \frac{\lambda}{n^r}$ with $0 < r < 1$, would help in constructing unbiased and consistent estimators. In our experiments, we did not find this choice to help.

### A.4.2  Differentially private Monte Carlo methods

Wang et al. [81] propose using Stochastic Gradient Langevin Dynamics [SGLD; 82] with a modified burn-in period and bounded step-size to provide DP sampling when the log-likelihood has bounded gradient. Li et al. [56] improve upon [81], taking advantage of the moments accountant [1] to allow for a larger step-size and faster mixing for non-convex target posteriors. Foulds et al. [27] extend their privatisation of sufficient statistics to a Gibbs sampling setting where the conditional posterior distribution for a Gibbs update is from the exponential family. Yıldırım and Ermiş [83] use the penalty algorithm which adds noise to the log of the Metropolis-Hastings acceptance probability. Heikkilä et al. [37] use Barker's acceptance test [8, 76] and provide RDP guarantees. Räisä et al. [73] derive DP-HMC also using the penalty algorithm. Zhang and Zhang [85] propose a random batch size implementation of Metropolis-Hastings for a general proposal distribution that takes advantage of the inherent randomness of Metropolis-Hastings and is asymptotically exact. Lastly, Awan and Rao [7] consider DP rejection sampling.

### A.5  Attack optimality

**Remark 1.**  *Let $p(\tilde{\theta}|D)$ be the density of the privacy mechanism—i.e the Laplace density for [18] or the posterior (i.e. (2) or (3)) for OPS. An attacker estimating $\mathcal{M}(\tilde{\theta}, D, D') = \frac{p(\tilde{\theta}|D')}{(p(\tilde{\theta}|D)+p(\tilde{\theta}|D'))}$ is Bayes optimal. For OPS, $\mathcal{M}(\tilde{\theta}, D, D') = \exp\{\ell(D_l'; f(\cdot;\tilde{\theta})) - \ell(D_l; \tilde{\theta})\} \int \exp\{\ell(D_l; f(\cdot;\theta)) - \ell(D_l'; f(\cdot;\theta))\}\pi(\theta|D)d\theta$ where $D, D'$ s.t. $D \setminus D' = \{D_l\}$ and $D' \setminus D = \{D_l'\}$ (see Appendix A.5).*

The privacy attacks outlined in Section 4 require the calculation of

$$\begin{aligned}
\mathcal{M}(\tilde{\theta}, D, D') := p(m = 1; \tilde{\theta}, D, D') &= p(\tilde{\theta}|D')/(p(\tilde{\theta}|D)) + p(\tilde{\theta}|D')) \\
&= 1/(p(\tilde{\theta}|D)/p(\tilde{\theta}|D') + 1)
\end{aligned}$$

by Bayes Theorem. For [18],

$$p(\tilde{\theta}|D) = \mathcal{L}\left(\hat{\theta}(D), \frac{2}{n\lambda\epsilon}\right),$$

where $\hat{\theta}(D)$ was defined in (1).

For the OPS methods, Minami et al. [66] and $\beta$D-Bayes, $p(\tilde{\theta}|D)$ is the posterior

$$p(\tilde{\theta}|D) = \pi^\ell(\tilde{\theta}|D) \propto \pi(\theta)\exp\left\{-\sum_{i=1}^n \ell(D_i;\theta)\right\}$$

where for [66] $\ell(D_i; f(\cdot;\theta)) = -w\log f(D_i;\theta)$, and for $\beta$D-Bayes $\ell(D_i; f(\cdot;\theta)) = \ell^{(\beta)}(D_i; f(\cdot;\theta))$ given in (4). Without loss of generality, index observations within $D$ and $D'$ such that $D \setminus D' = \{D_l\}$ and $D' \setminus D = \{D'_l\}$. Then,

$$\frac{\pi^\ell(\tilde{\theta}|D)}{\pi^\ell(\tilde{\theta}|D')} = \frac{\pi(\tilde{\theta})\exp\{-\sum_{i=1}^n \ell(D_i; f(\cdot;\tilde{\theta}))\}}{\int \pi(\theta)\exp\{-\sum_{i=1}^n \ell(D_i; f(\cdot;\theta))\}d\theta} \Big/ \frac{\pi(\tilde{\theta})\exp\{-\sum_{i=1}^n \ell(D'_i; f(\cdot;\tilde{\theta}))\}}{\int \pi(\theta)\exp\{-\sum_{i=1}^n \ell(D'_i; f(\cdot;\theta))\}d\theta}$$

$$= \exp\{\ell(D'_l; f(\cdot;\tilde{\theta})) - \ell(D_l; f(\cdot;\tilde{\theta}))\} \int \frac{\pi(\theta)\exp\{-\sum_{i=1}^n \ell(D'_i; f(\cdot;\theta))\}}{\int \pi(\theta)\exp\{-\sum_{i=1}^n \ell(D_i; f(\cdot;\theta))\}d\theta}d\theta$$

$$= \exp\{\ell(D'_l; f(\cdot;\tilde{\theta})) - \ell(D_l;\tilde{\theta})\}$$
$$\int \exp\{\ell(D_l;\theta) - \ell(D'_l; f(\cdot;\theta))\}\frac{\pi(\theta)\exp\{-\sum_{i=1}^n \ell(D_i; f(\cdot;\theta))\}}{\int \pi(\theta)\exp\{-\sum_{i=1}^n \ell(D_i; f(\cdot;\theta))\}d\theta}d\theta$$

$$= \exp\{\ell(D'_l; f(\cdot;\tilde{\theta})) - \ell(D_l; f(\cdot;\tilde{\theta}))\} \int \exp\{\ell(D_l; f(\cdot;\theta)) - \ell(D'_l; f(\cdot;\theta))\}\pi(\theta|D)d\theta$$

$$\approx \exp\{\ell(D'_l; f(\cdot;\tilde{\theta})) - \ell(D_l; f(\cdot;\tilde{\theta}))\}\frac{1}{N}\sum_{j}^{N}\exp\{\ell(D_l; f(\cdot;\theta_j)) - \ell(D'_l; f(\cdot;\theta_j))\},$$

where $\{\theta_j\}_{j=1}^N \sim \pi(\theta|D)$. The adversary only needs to sample from the posterior based on dataset $D$ to be able to estimate $\mathcal{M}(\tilde{\theta}, D, D')$ for all $D'$ differing from $D$ in only one index $l$.

## Appendix B  Additional experimental details and results

**Datasets** Please see the following list for the name of the targets in the prediction tasks of the UCI data:

- abalone - ring class (threshold 10)
- adult - income class
- bank - authenticity
- boston - housing prices
- kin8mm - distance of the end-effector from a target
- naval - GT Turbine decay state coefficient
- wine - quality
- vertebral - disease (normal/abnormal)

We used all remaining features in the data sets as predictors.

**Bayesian models** In the case of logistic regression, we assume a normal prior with mean 0 and variance 9 on the regression coefficient. In the case of linear regression, we again assume a normal prior with mean 0 and variance $9 \cdot \sigma^2$ on the regression coefficient. Further, we assume $\sigma^2$ follows an inverse gamma distribution a-priori with shape and scale of 1, with a lower bound of $0.01$. In the case of neural network classification, we assume normal priors with mean 0 and variance 1 on the weight and bias parameters of a one-hidden-layer neural network with 10 hidden nodes. In the case of neural network regression, the same assumptions hold and we additionally assume $\sigma^2$ follows an inverse gamma distribution a-priori with shape and scale of 1, with a lower bound of $0.01$.

**MCMC sampling**   Unless otherwise specified, we choose $d = 2$ in the simulated experiments. The MCMC methods are run for 1000 warm-up steps, and 100 iterations. The DP sample is sampled uniformly at random from the last 100 iterations.

**Neural network training**   DPSGD is run for $14 + \lfloor \epsilon \rfloor$ epochs, with clipping norm 1, batch size 100, and learning rate of $10^{-2}$. All other implementation details can be found at `https://github.com/sghalebikesabi/beta-bayes-ops`. We tuned the learning rate and the batch size on validation data sets when training with DPSGD. We use the same parameters for training with SGD to show how much performance is lost through the privatisation of the training procedure. The rule for the number of epochs was found by tuning the number of epochs for $\epsilon = 1$ within $\{5, 10, 15, 20\}$ and trying different parameters for the slope of the linear increase in $\epsilon$ ($\{0.5, 1, 2\}$) following observations from [20]. The batch size was tuned within $\{50, 100, 200\}$, the learning rate within $\{10^{-3}, 10^{-2}, 10^{-1}, 1\}$, and the clipping norm within $\{0.8, 1, 1.2\}$.

**Additional plots for logistic regression**   Figure 6 compares the DP method for logistic regression individually with their non-private counterparts. For the $\beta$D-Bayes the non-private estimate was the posterior mean (PM) while for [18] it was the regularised empirical risk minimiser $\hat{\theta}(D)$ without the added noise. As the number of observations $n$ increases the DP-estimates from $\beta$D-Bayes and [18] with $\lambda = 1/9$ approach the unprivatised estimates. This is not the case for [18] with $\lambda = 1/9n$.

Figure 7 and 8 consider the correct sign accuracy (CSA) of the DP methods for logistic regression. The CSA is the proportion of the time the DP-parameter estimates from the different methods agree with those of a non-private baselines. The plot demonstrates that the estimates from $\beta$D-Bayes OPS agree with the sign of the non-private method more of the time than the other methods.

**Neural network classification**   Similarly to neural network regression, we can also use $\beta$D-Bayes for DP neural network classification. For simulated data, Figure 9 shows the for $\epsilon > 0.2$ the $\beta$D-Bayes generally achieves higher test set ROC-AUC than DPSGD across all sample sizes. In the real data we see that for $\epsilon = 0.5$ $\beta$D-Bayes and DPSGD perform similarly, on `abalone` the DPSGD generally achieves higher ROC-AUC whereas on `bank` the $\beta$D-Bayes generally performs slightly better. However, for $\epsilon = 5$, we see that while the DPSGD sometimes achieves higher ROC-AUC when $n$ is small, as $n$ increases the $\beta$D-Bayes achieves the highest ROC-AUC for all datasets, once again illustrating the consistency of our method.

**Sensitivity in number of features**   Figure 10 investigates the quality of the DP estimation methods considered in this paper as the number of features grows. The top panel considers parameter RMSE of the data generating parameter $\theta$ (divided by the number of dimensions of $\theta$) in logistic regression. Here even the non-DP methods increase as the dimension of the parameter space increases as more parameters are estimated from the same number of observations. While the $\beta$D-Bayes performs the best, it appears to scale similarly with dimension as [18] and [66]. The middle panel considers test set predictive RMSE in neural network regression. More features provide a more complex model which should decrease RMSE. We see that the $\beta$D-Bayes achieves lower RMSE than DPSGD and $\beta$D-Bayes appears to scale better with dimension as the improvement made by $\beta$D-Bayes improves as the dimension of the feature space increases. The bottom panel considered test set ROC-AUC for neural network classification. Again, more features provide a more complex classifier improving test set ROC-AUC. For $\epsilon > 0.2$ $\beta$D-Bayes achieves a higher ROC-AUC than DPSGD and the improvement increases with the number of features.

**Membership inference attacks**   For $\epsilon \in \{0.2, 1, 2, 7, 10, 20\}$, we run 10,000 rounds of the attack presented in Section 4. Figure 11 presents an estimated lower bound on $\epsilon$, given the false positive and negative rates of the attacks [43], for each method with its RMSE for the data generating $\theta$. Note that these lower bounds are unrealistic for $\epsilon < 1$. We see that, for any RMSE value, $\beta$D-Bayes achieves a lower practical bound on $\epsilon$ than [18], which gives exact privacy guarantees. For fixed lower bound, we see that $\beta$D-Bayes achieves the smallest RMSE.

**Compute**   While the final experimental results can be run within approximately two hours on a single Intel(R) Xeon(R) Gold 5118 CPU @ 2.30GHz core, the complete compute needed for the final results, debugging runs, and sweeps amounts to around 11 days.

**Licenses**   The UCI data sets are licensed under Creative Commons Attribution 4.0 International license (CC BY 4.0).

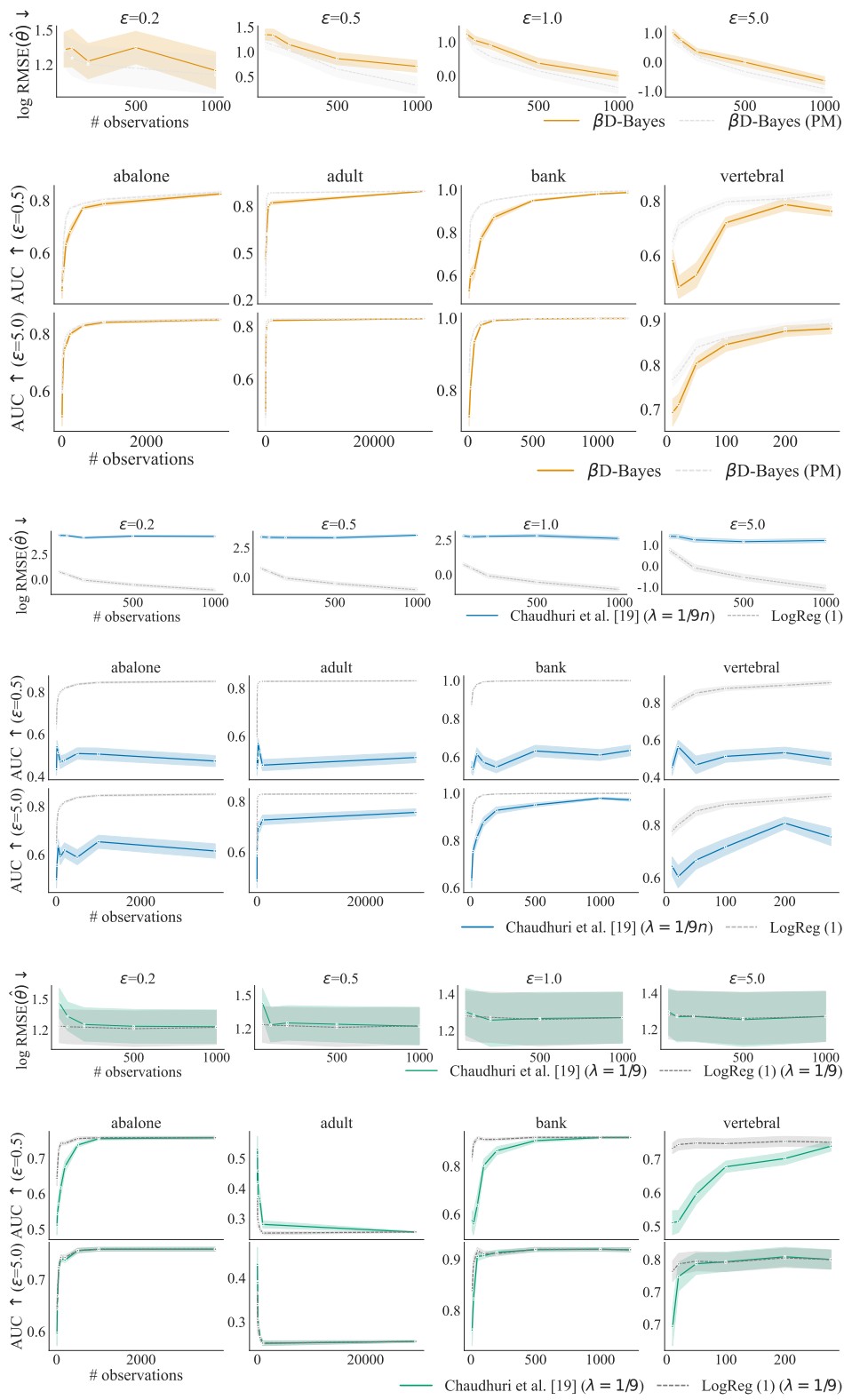

Figure 6: Parameter log RMSE and test-set ROC-AUC of DP estimation for logistic regression as the number of observations $n$ increases on simulated (first row of each method) and UCI data (second and third row of each method). PM stands for posterior mean which was estimated over 20 samples. This is a non-private baseline, i.e. the point estimate you would release if privacy were not an issue.

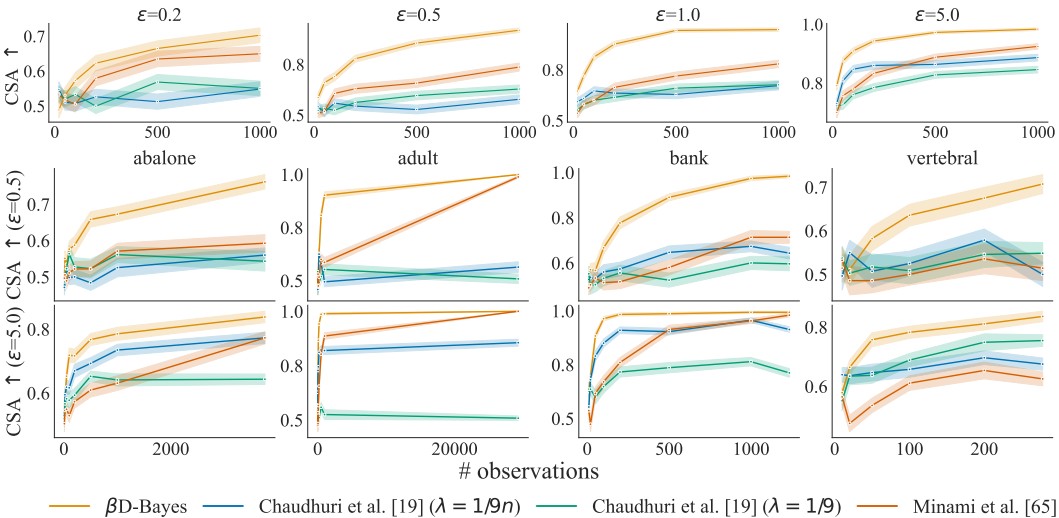

Figure 7: Correct sign accuracy (CSA) for simulated experiments (first row) and UCI data sets (second and third rows). For CSA, we estimate the number of coefficients in the logistic regression that have the same sign as the true coefficient on the simulated experiment or the non-DP coefficient estimated on the UCI data set with $L_2$-regularised logistic regression as implemented with default parameters in |sklearn|.

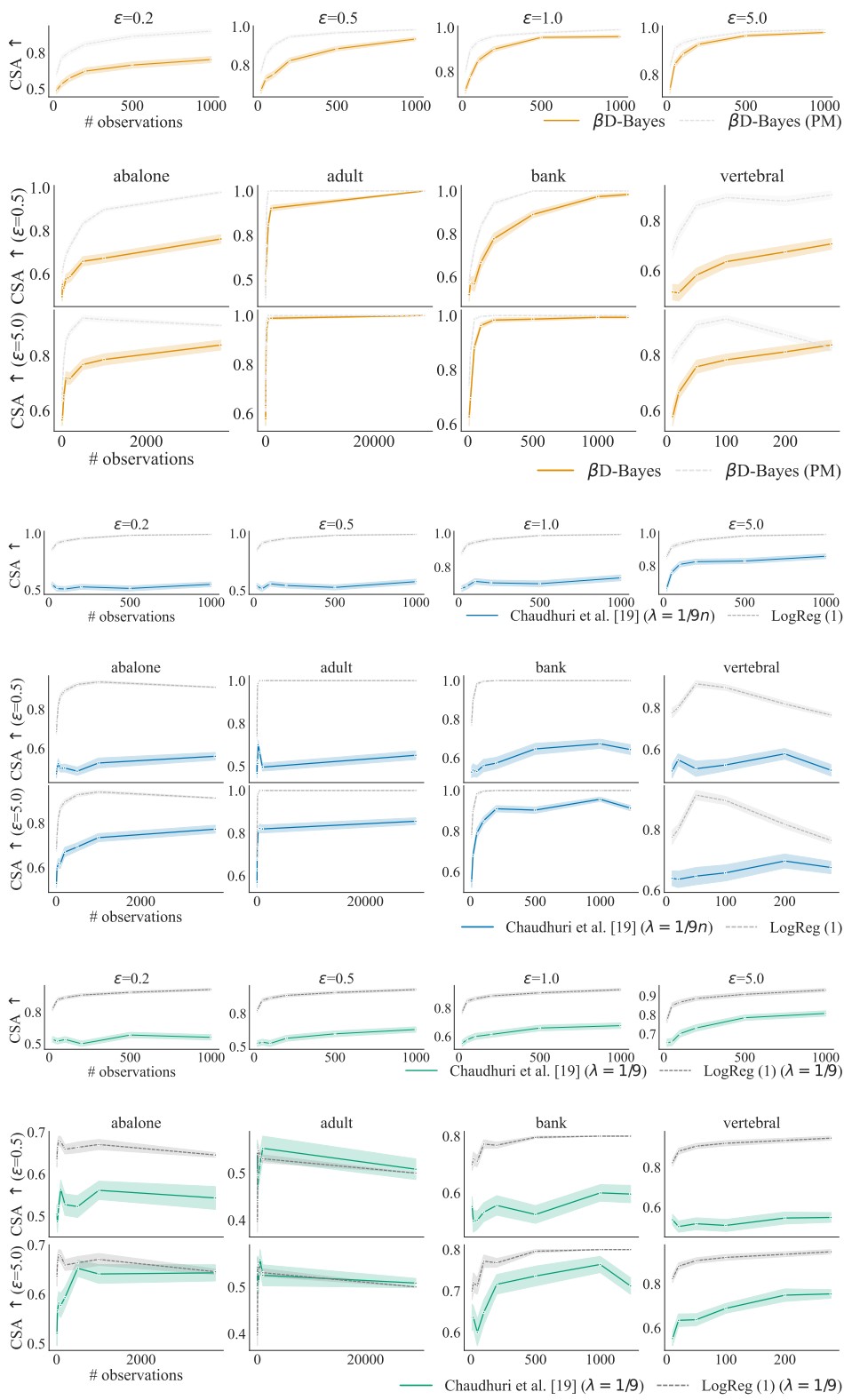

Figure 8: CSA for simulated experiments (first row of each method) and UCI data sets (second row of each method). PM stands for posterior mean which was estimated over 20 samples. This is a non-private baseline, i.e. the point estimate you would release if privacy were not an issue.

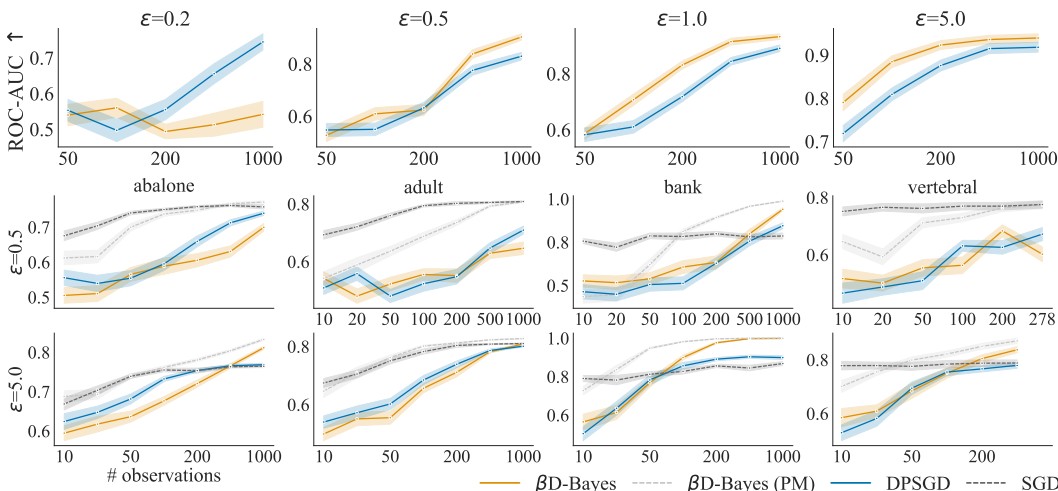

Figure 9: Test set predictive ROC-AUC of DP estimation for neural network classification as the number of observations $n$ increases on simulated and UCI data.

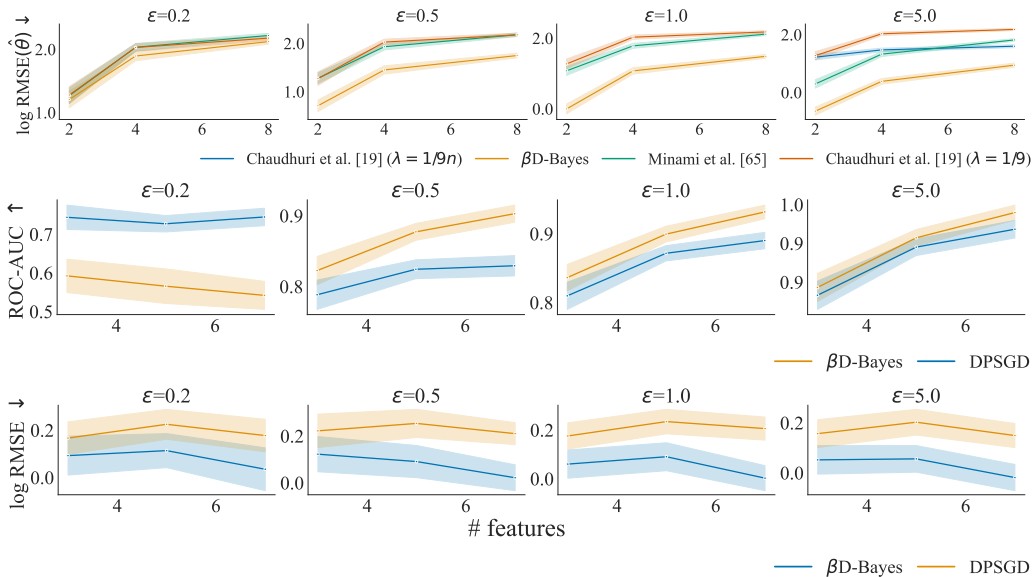

Figure 10: Parameter log RMSE of DP logistic regression (**first row**), test set predictive log RMSE of DP neural network classification (**second row**), and test set ROC-AUC of DP neural network regression (**third row**) as the number of features $d$ increases on simulated data with $n = 1000$.

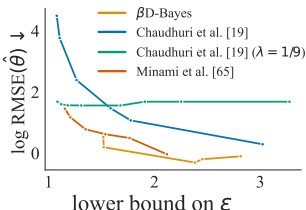

Figure 11: Lower bound on $\epsilon$ against log RMSE. Points correspond to values of $\epsilon$.

