# Appendix A  Definitions, proofs, and related work

Here, we provide missing definitions of the KLD and the $\beta$-divergence.

**Definition 2** (Kullback-Leibler divergence [53]). *The* KLD *between probability densities $g(\cdot)$ and $f(\cdot)$ is given by*

$$\text{KLD}(g||f) = \int g(x) \log \frac{g(x)}{f(x)} dx.$$

**Definition 3** ($\beta$-divergence [9, 63]). *The $\beta$-divergence is defined as*

$$D_B^{(\beta)}(g||f) = \frac{1}{\beta(\beta-1)} \int g(x)^\beta dx + \frac{1}{\beta} \int f(x)^\beta dx - \frac{1}{\beta-1} \int g(x) f(x)^{\beta-1} dx,$$

*where $\beta \in \mathbb{R} \backslash \{0, 1\}$. $D_B^{(\beta)}$ is a member of the Bregman-divergence family [16] with $\psi(t) = \frac{1}{\beta(\beta-1)} t^\beta$. When $\beta \to 1$, $D_B^{(1)}(g(x)||f(x)) \to \text{KLD}(g(x)||f(x))$.*

The $\beta$-divergence has often been referred to as the *density-power divergence* in the statistics literature [9] where it is often parameterised with $\beta_{DPD} = \beta - 1$.

Intuition for how $\beta$D-Bayes provides DP estimation is provided in Figure 5 which shows the divergence between the posterior before and after adding an observation $y$ that is $|y - \mu|$ standard deviations away from the posterior mean $\mu$ when updating using a Gaussian distribution under KLD-Bayes and $\beta$D-Bayes. The influence of observations under KLD-Bayes is steadily increasing, making the posterior sensitive to extreme observations and therefore leaking their information. Under $\beta$D-Bayes, the influence initially increases before being maximised at a point depending on the value of $\beta$, before decreasing to 0. Therefore, each observation has bounded influence on the posterior, allowing for DP estimation.

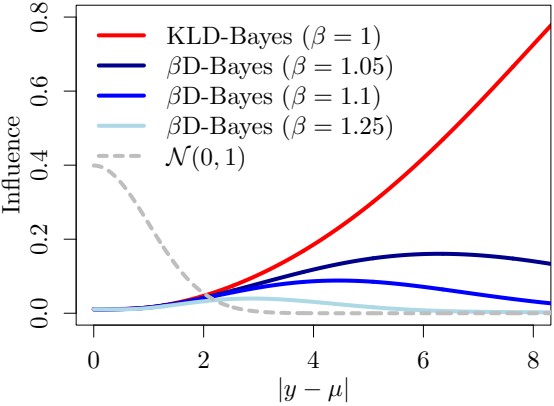

Figure 5: The influence of adding an observation $y$ with $|y - \mu|$ on the posterior conditioned on a sample of 1000 points from a $\mathcal{N}(0, 1)$ when fitting a $\mathcal{N}(\mu, \sigma^2)$.

## A.1  Bernstein-von Mises theorem for $\beta$D-Bayes

The general Bernstein-von Mises theorem for generalised posteriors [Theorem 4; 64] can be applied to the $\beta$D-Bayes posterior to show that

$$\int \left| \tilde{\pi}^{(\beta)}(\phi) - \mathcal{N}\left(\phi; 0, (H_0^{(\beta)})^{-1}\right) \right| d\phi \xrightarrow[n \to \infty]{} 0 \tag{5}$$

where $\tilde{\pi}^{(\beta)}$ denotes the density of $\sqrt{n}(\tilde{\theta} - \hat{\theta}_n^{(\beta)})$ when $\tilde{\theta} \sim \pi^{(\beta)}(\cdot; D)$, $\mathcal{N}(x; \mu, \sigma^2)$ denotes the normal distribution with mean $\mu$ and variance $\sigma^2$, and

$$\hat{\theta}_n^{(\beta)} := \arg\min_{\theta \in \Theta} \sum_{i=1}^n \ell^{(\beta)}(D_i, f(\cdot; \theta)), \quad \theta_0^{(\beta)} := \arg\min_{\theta \in \Theta} \mathbb{E}_g \left[ \ell^{(\beta)}(D, f(\cdot; \theta)) \right]$$

$$H_0^{(\beta)} := \left( \frac{\partial}{\partial \theta_i \partial \theta_j} \mathbb{E}_D \left[ \ell^{(\beta)}(D, f(\cdot; \theta_0^{(\beta)})) \right] \right)_{i,j}.$$

614  That is to show that the $\beta$D-Bayes posterior converges to a Gaussian distribution centered around the
615  $\beta$D minimising parameter $\theta_0^{(\beta)}$ in total variation distance.

## A.2  Proofs

### A.2.1  Proof of Lemma 1

618  **Lemma 1** (Bounded sensitivity of the $\beta$D-Bayes loss)**.** *Under Condition 1 the sensitivity of the*
619  $\beta$D-*Bayes-loss for any* $\beta > 1$ *is* $\left| \ell^{(\beta)}(D, f(\cdot; \theta)) - \ell^{(\beta)}(D', f(\cdot; \theta)) \right| \leq \frac{M^{\beta-1}}{\beta-1}.$

620  *Proof.* By (4), for $\beta > 1$

$$\left| \ell^{(\beta)}(D, f(\cdot; \theta)) - \ell^{(\beta)}(D', f(\cdot; \theta)) \right| = \frac{1}{\beta-1} \left( f(D'; \theta)^{\beta-1} - f(D; \theta)^{\beta-1} \right)$$

$$\leq \max_D \frac{1}{\beta-1} f(D; \theta)^{\beta-1})$$

$$\leq \frac{M^{\beta-1}}{\beta-1}$$

621  $\qquad\qquad\qquad\qquad\qquad\qquad\qquad\qquad\qquad\qquad\qquad\qquad\qquad\qquad\qquad\qquad\qquad\qquad\qquad\qquad$ $\square$

### A.2.2  Proof of Theorem 1

623  **Theorem 1** (Differential privacy of the $\beta$D-Bayes posterior)**.** *Under Condition 1, a draw* $\tilde{\theta}$ *from the*
624  $\beta$D-*Bayes posterior* $\pi^{(\beta)}(\theta|D)$ *in* (3) *is* $\left( \frac{2M^{\beta-1}}{\beta-1}, 0 \right)$-*differentially private.*

625  *Proof.* Define $D = \{D_1, \ldots, D_n\}$, $D' = \{D_1', \ldots, D_n'\}$ and let $j$ be the index such that $D_j \neq D_j'$
626  with $D_j = D_i'$ for all $i \neq j$. Firstly, the normalising constant of the $\beta$D-Bayes posterior combining
627  (3) with (4) is

$$P^\ell(D) := \int \pi(\theta) \exp\left( -w \sum_{i=1}^n \ell\{\theta, D_i\} \right) d\theta \tag{6}$$

628  Then,

$$\log \frac{\pi^{(\beta)}(\theta|D)}{\pi^{(\beta)}(\theta|D')} = \sum_{i=1}^n \ell^{(\beta)}(D_i', f(\cdot; \theta)) - \sum_{i=1}^n \ell^{(\beta)}(D_i, f(\cdot; \theta)) + \log \frac{P^{(\beta)}(D')}{P^{(\beta)}(D)}$$

$$= \ell^{(\beta)}(D_j'; f(\cdot; \theta)) - \ell^{(\beta)}(D_j; f(\cdot; \theta)) + \log \frac{P^{(\beta)}(D')}{P^{(\beta)}(D)}$$

629  where $P^{(\beta)}(D')$ is the normaliser of the general Bayesian posterior defined in (3).

630  Now, by Condition 1 and Lemma 1,

$$\ell^{(\beta)}(D_j'; f(\cdot; \theta)) - \ell^{(\beta)}(D_j; f(\cdot; \theta)) \leq \frac{M^{\beta-1}}{\beta-1},$$

631  and

$$P^{(\beta)}(D') = \int \exp\left\{ -\sum_{i=1}^n \ell^{(\beta)}(D_i', f(\cdot; \theta)) \right\} \pi(\theta) d\theta$$

$$= \int \exp\left\{ \ell^{(\beta)}(D_j, f(\cdot; \theta)) - \ell^{(\beta)}(D_j', f(\cdot; \theta)) - \sum_{i=1}^n \ell^{(\beta)}(D_i, f(\cdot; \theta)) \right\} \pi(\theta) d\theta$$

$$= \exp\left\{ \frac{M^{\beta-1}}{\beta-1} \right\} \int \exp\left\{ -\sum_{i=1}^n \ell^{(\beta)}(D_i, f(\cdot; \theta)) \right\} \pi(\theta) d\theta,$$

632  which combined provides that

$$\log \frac{\pi(\theta|D)}{\pi(\theta|D')} \leq 2 \frac{M^{\beta-1}}{\beta-1}.$$

633  $\qquad\qquad\qquad\qquad\qquad\qquad\qquad\qquad\qquad\qquad\qquad\qquad\qquad\qquad\qquad\qquad\qquad\qquad\qquad\qquad$ $\square$

### A.2.3 Proof of Theorem 2

**Theorem 2** (Consistency of $\beta$D-Bayes sampling). *Under the conditions of Theorem 4 of [64],*

*1. a posterior sample $\tilde{\theta} \sim \pi^{(\beta)}(\theta|D)$ is a consistent estimator of $\theta_0^{(\beta)}$.*

*2. if data $D_1, \ldots, D_n \sim g(\cdot)$ were generated such that there exists $\theta_0$ with $g(D) = f(D; \theta_0)$, then*
   *$\tilde{\theta} \sim \pi^{(\beta)}(\theta|D)$ for all $1 \leq \beta \leq \infty$ is consistent for $\theta_0$.*

*Proof.* For part 1), define $B_r(x_0) = \{x \in \mathbb{R}^p : |x - x_0| < r\}$. Theorem 4 of [64] applied to $\beta$D-Bayes posterior proves that

$$\int_{B_\varepsilon(\theta_0^{(\beta)})} \pi^{(\beta)}(\theta|D)d\theta \underset{n \to \infty}{\longrightarrow} 1$$

for all $\varepsilon > 0$. This is enough to show that for $\tilde{\theta} \sim \pi^{(\beta)}(\theta|D) \to \theta_0^{(\beta)}$ in probability.

For part 2), note that if $g(D) = f(D; \theta_0)$, then for all $1 \leq \beta \leq \infty$

$$\theta_0^{(\beta)} := \underset{\theta \in \Theta}{\arg\min} \, \mathbb{E}_g \left[ \ell^{(\beta)}(D; f(\cdot; \theta)) \right]$$
$$= \underset{\theta \in \Theta}{\arg\min} \, D_B^{(\beta)}(g||f(\cdot; \theta))$$
$$= \theta_0.$$

$\square$

### A.2.4 Proof of Proposition 1

**Proposition 1** (Asymptotic efficiency). *Under the conditions of Theorem 4 of [64], $\tilde{\theta} \sim \pi^{(\beta)}(\theta|x)$ is asymptotically distributed as $\sqrt{n}(\tilde{\theta} - \theta_0^{(\beta)}) \overset{weakly}{\longrightarrow} \mathcal{N}(0, (H_0^{(\beta)})^{-1}K_0^{(\beta)}(H_0^{(\beta)})^{-1} + (H_0^{(\beta)})^{-1})$, where $K_0^{(\beta)}$ and $H_0^{(\beta)}$ are defined in Appendix A.1.*

*Proof.* Let $\tilde{\theta} \sim \pi^{(\beta)}(\theta|D)$. By the Bernstein-von Mises theorem [64] applied to $\beta$D-Bayes in (5),

$$\sqrt{n}(\tilde{\theta} - \hat{\theta}_n^{(\beta)}) \to \mathcal{N}(0, (H_0^{(\beta)})^{-1}).$$

By the asymptotic normality of $\hat{\theta}_n^{(\beta)}$ [10], we have that

$$\sqrt{n}(\hat{\theta}_n^{(\beta)} - \theta_0^{(\beta)}) \to^D \mathcal{N}(0, (H_0^{(\beta)})^{-1}K_0^{(\beta)}(H_0^{(\beta)})^{-1})$$

for $K_0 := \left( \frac{\partial}{\partial\theta_i}\mathbb{E}_D \left[ \ell^{(\beta)}(D; f(\cdot; \theta_0^{(\beta)})) \right] \frac{\partial}{\partial\theta_j}\mathbb{E}_D \left[ \ell^{(\beta)}(D; f(\cdot; \theta_0^{(\beta)})) \right] \right)_{i,j}$. The result then comes from the asymptotic independence of $\tilde{\theta} - \hat{\theta}_n^{(\beta)}$ and $\hat{\theta}_n^{(\beta)}$ [see e.g. 80]                                $\square$

### A.2.5 Proof of Proposition 2

**Proposition 2** (DP-MCMC methods for the $\beta$D-Bayes-Posterior). *Under Condition 1, the penalty algorithm of [Algorithm 1; 82], DP-HMC of [Algorithm 1; 72] and DP-Fast MH of [Algorithm 2; 84] and under further Condition 2 DP-SGLD of [Algorithm 1; 56] can be used to produce $(\epsilon, \delta)$-DP estimation from the $\beta$D-Bayes posterior with $\delta > 0$ without requiring the clipping of any gradients.*

**Condition 2** (Boundedness of the model density/mass function gradient). *The model density or mass function $f(\cdot; \theta)$ is such that there exists $0 < G^{(\beta)} < \infty$ such that $\left| \nabla_\theta f(D; \theta) \times f(D; \theta)^{\beta-2} \right| \leq G^{(\beta)}, \forall \theta \in \Theta$.*

*Proof.* Algorithm 1 of [82], Algorithm 1 of [72] and Algorithm 2 of [84] requires a posterior whose log-likelihood has bounded sensitivity. For $\beta$D-Bayes posterior, this requires $\beta$D-Bayes-loss has bounded sensitvity which is provided by Condition 1 and Lemma 1.

Algorithm 1 of [56] requires a posterior whose log-likelihood has bounded gradient. For $\beta$D-Bayes posterior, this requires $\beta$D-Bayes-loss to have bounded gradient:

$$\begin{aligned}
|\nabla_\theta \ell^{(\beta)}(D;\theta)| &= \nabla_\theta f(D;\theta) \times f(D;\theta)^{\beta-2} - \int \nabla_\theta f(D;\theta) \times f(D;\theta)^{\beta-1} dD \\
&= \nabla_\theta f(D;\theta) \times f(D;\theta)^{\beta-2} - \int \nabla_\theta f(D;\theta) \times f(D;\theta)^{\beta-2} \times f(D;\theta) dD \\
&\leq \max\{G^{(\beta)}, G^{(\beta)}M\},
\end{aligned}$$

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

 use $\beta$D-Bayes for neural network classification. As we see in Figure 6, $\beta$D-Bayes regularly outperforms DPSGD for $\epsilon > 0.2$ on simulated and real data, except on `abalone`.

**Sensitivity in number of features**   Please refer to Figure 7 for the sensitivity of the private methods w.r.t. the number of features in the data set. We see that the RMSE of the data generating parameter $\theta$ (divided by the number of dimensions of $\theta$) increases. The reason for this is two-fold: 1) The methods of [19] and [65] provide their privacy guarantees w.r.t. the number of features. While more noise has to be added for [19], the influence of the prior increases for [65] when the number of features increases for a fixed privacy budget. 2) A single sample from a posterior is of higher variance the higher-dimensional the posterior is, negatively influencing OPS methods such as [65] and $\beta$D-Bayes.

**Membership inference attacks**   For $\epsilon \in \{0.2, 1, 2, 7, 10, 20\}$, we run 10,000 rounds of the attack presented in Section 4. In Figure 8, we use the approach presented by [44] to estimate a lower bound on $\epsilon$ given the false positive and negative rates of the attacks. Note that these lower bounds are unrealistic for $\epsilon < 1$. We see that, for any RMSE value, $\beta$D-Bayes achieves a lower practical bound on $\epsilon$ than [19], which gives exact privacy guarantees.

**Compute**   While the final experimental results can be run within approximately two hours on a single Intel(R) Xeon(R) Gold 5118 CPU @ 2.30GHz core, the complete compute needed for the final results, debugging runs, and sweeps amounts to around 11 days.

**Licenses**   The UCI data sets are licensed under Creative Commons Attribution 4.0 International license (CC BY 4.0).

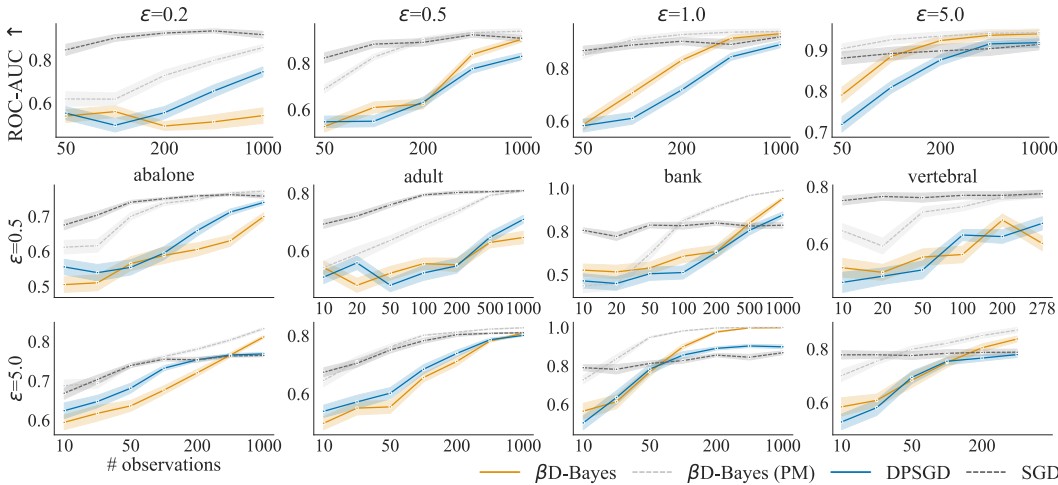

Figure 6: Test set predictive ROC-AUC of DP estimation for neural network classification as the number of observations $n$ increases on simulated and UCI data.

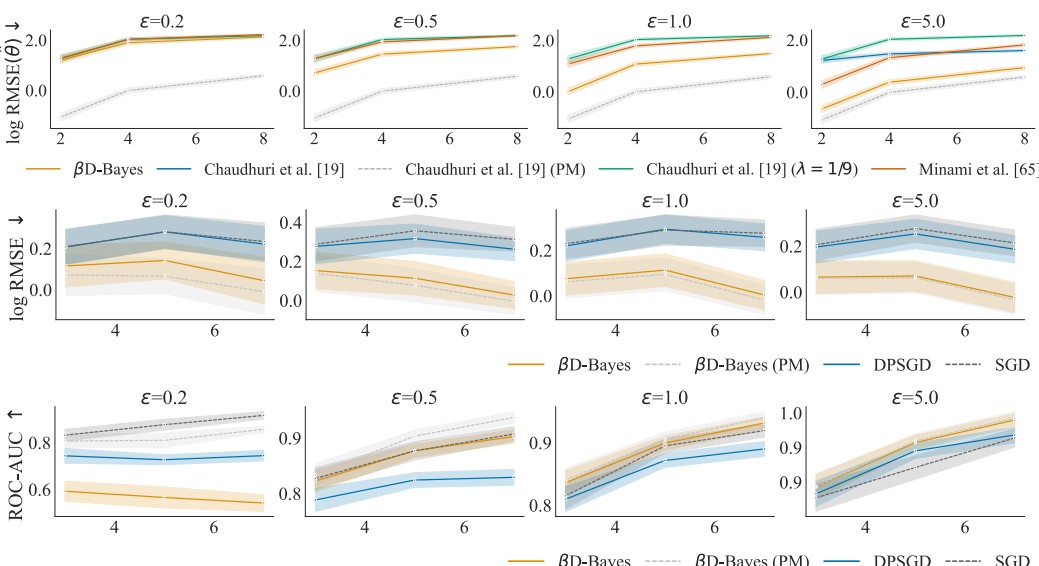

Figure 7: Parameter log RMSE of DP logistic regression (**first row**), test set predictive log RMSE of DP neural network regression (**second row**), and test set ROC-AUC of DP neural network classification (**third row**) as the number of features $d$ increases on simulated data with $n = 1000$.

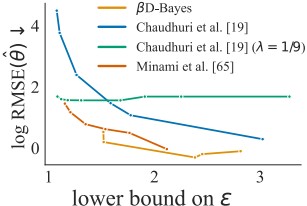

Figure 8: Lower bound on $\epsilon$ against log RMSE. Points correspond to values of $\epsilon$.