# OpenReview forum: "Differentially Private Statistical Inference through $\beta$-Divergence One Posterior Sampling"
_NeurIPS.cc/2023/Conference — NeurIPS 2023 poster_

### Official Review · Reviewer_D3oa · 2023-06-27

**Soundness:** 4 excellent
**Presentation:** 3 good
**Contribution:** 3 good
**Rating:** 6
**Confidence:** 2

**Summary:**

This paper combines OPS  (one posterior sampling) with bounded beta-divergence, resulting in a DP mechanism that applies to a general class of inference models. This approach bounds the sensitivity of the procedure without bounding the feature space or statistical functionals thereby improving on previous approaches. The paper provides empirical evidence for the performance of the approach against relevant baselines and on multiple different data sets.

**Strengths:**

The paper is well motivated, I agree that overcoming the limitations of sensitivity bounding is an important direction for work on differentially private inference. The theoretical analysis is careful and precise. Table 1 is very helpful for baselining the approach and clarifying its novelty.



**Weaknesses:**

The presentation of section 3 is extremely dense and jargon-y,  with very little intuition given for the key formulas (e.g. equation 4). The paper would be significantly easier to follow to those unfamiliar with this class of Bayesian inference algorithms (which are fairly esoteric as far as I gather) if some additional intuition was included.

Figure 2 is very hard to read given the amount of information being conveyed and the size (e.g. the red and orange colors are similar, what is PM?). From looking at the figure, it is also not obvious to me that the proposed method is better. Perhaps extrapolating the line out in n, that would be the case, but then these experiments should be run with larger sample sizes. If the reason for not doing so is the computational infeasibility then this should be noted as a limitation of the method.

The description of the experiments was lacking clarity in my opinion. What is the prediction goal of training these models? what information is contained in these datasets? how many covariates were used? how/why were hyper-parameters chosen? These details should be included in the main body of the paper. I would be more convinced if the experiment could show that the proposed approach replicated a finding without DP with lower error than alternatives.

**Questions:**


How can this approach be used for uncertainty quantification over the parameters or posterior predictive? In approaches where the exact posterior is targeted, this is obvious but it is less clear with this approach. If this requires additional privacy budget, this should be noted.



**Limitations:**

The authors should address the computational demandingness of this approach compared to alternative approaches. How large of a model/dataset could this realistically be run on?

---

> ### Author Rebuttal · Authors · 2023-08-09
>
> Thank you for acknowledging the importance of our contribution to overcoming the limitations of sensitivity bounding. We address your questions and concerns below.
> ### Improving the presentation of Sec. 3 and providing more intuition
> The most important points for readers unfamiliar with Bayesian and general Bayesian inference to understand are:
> + The Gibbs posterior used by Wang [80] and Minami [65] generalises the standard Bayesian posterior to $\pi(\theta|D) \propto \pi(\theta)\exp(\sum_{i=1}^n w\log f(D_i; \theta))$. The parameter $w > 0$ provides some flexibility in adapting to the sensitivity of the posterior. However, bounding the sensitivity of $\log f(D_i; \theta)$ is difficult outside very simple problems.
> + The generalised Bayesian posterior, Bissiri [15], generalises this further to $\pi(\theta|D) \propto \pi(\theta)\exp(-\sum_{i=1}^n \ell(D_i; \theta))$ (Eq. 3) providing the flexibility to choose a loss with bounded sensitivity.
> + We chose the beta-divergence loss (Eq. 4), which still depends on the model $f(\cdot;\theta)$, and allows data generating parameters to be learned unbiasedly, while also having bounded sensitivity.
> For improving clarity and intuition, we propose the following changes:
> +ll 123 "Note $w = 1$ recovers the standard posterior and $w\neq 1$ provides flexibility to adapt the posterior the level of privacy required.’’
> + ll 153 "OPS has struggled as a general purpose tool for DP estimation as bounding the sensitivity of $\log f(x;\theta)$ is difficult.’’.
> + ll 155 “high posterior density is assigns to parameters that achieved small loss on the data.’’
> + ll 156 "The Gibbs posterior in (2) is recovered using the weighted negative log-likelihood … and the standard posterior for $w=1$.’’
> + ll 158 "The framework of Bissiri et al. [15] provides the flexibility to choose a loss function with bounded sensitivity. An alternative loss function…’’
> + ll 161 "the first term in (4) contains the negative likelihood, therefore parameters that make the data likely achieve low loss. However, it is raised to the power $\beta-1 > 1$ prescribing relatively smaller loss to observations unlikely under that parameter than the log-likelihood. A key feature of (4) is that while $lim_{f\rightarrow 0} -\log f = \infty$, $lim_{f\rightarrow 0} -\frac{f^{\beta-1}}{\beta-1} = 0$ for $\beta > 1$. The second integral term only depends on the parameters and ensures the $\beta$D loss can learn the data generating parameters.’’
> + ll 163 "$\beta = 1$ recovers the negative log-likelihood.’’
> + ll 180-183 “Rather than bounding $log f(\cdot;\theta)$, we replace it in (3) with the $\beta$D loss from (4) which is naturally bounded when the density is bounded.’’
> We will also use more subsections to improve the readability.
> ### Improving the description of the experiments
> Thank you. Appendix B will be updated to contain the following information for each dataset/experiment
> + What was the response
> + What were the predictors
> + Specifications of priors, regularisers and optimisation parameters
> Most hyperparameters resulted directly from the DP requirement.
> ### Extensions to uncertainty quantification
> In this work, we are not advocating Bayesianism as the correct paradigm for inference, but as a convenient way to release DP parameter estimates as an alternative to adding noise to the empirical risk minimizer. We demonstrate improved performance and flexibility.
>
> If uncertainty quantification is required, the privacy budget can be split and more than one sample can be released from a posterior with higher $\beta$. Interesting further work could look at the utility/privacy trade-off here.
>
> To Sec. 6 ll 369 we will add "Extensions of this work could consider the benefits of dividing the privacy budget to allow for the release of more than one sample paving the way for parameter inference as well as estimation.’’
> ### Addressing the computational demandingness of this approach compared to alternative approaches
> In Sec. 4 and 6 we identify the limitation that our approach relies on a perfect sample from the posterior. This is shared by the Gibbs OPS approaches of Wang [80] and Minami [65], but not by the approaches of Chaudhuri [19] or DPSGD [1]. These rely on techniques from optimisation that are known to scale better in the number of parameters.
> Obtaining posterior samples is left for future work. In Sec. 4 we discuss how the $\beta$D-Bayes posterior is naturally suitable for application to the DP-MCMC literature. A great deal of research has gone into scaling MCMC methods for logistic regression and there are tailored MCMC methods for neural networks as well. Finally, these procedures cannot be repeated, as repeated estimation would leak information, and only require one posterior sample (after reaching stationarity).
> We hope that the considerable improvements in performance make the increased computational costs worthwhile. Our experiments show that $\beta$D-Bayes OPS outperforms Chaudhuri [19] and DPSGD [1] as well as the Gibbs OPS methods. We hope that our method encourages further research into computational procedures for these models, including scaling MCMC to large neural networks or alternatively DP variational Bayes methods to approximate the $\beta$D-Bayes posterior [41, 45, 70].
> To Sec. 6 l. 377 we will add:
> > A further limitation of OPS is the computational burden required to produce posterior samples, particularly in larger neural networks with many parameters. However, we argue that the improved performance justified such a cost and is mitigated by the fact that such inference can only be run once to avoid leaking privacy and that only one posterior sample is required. We hope that the performance of our method encourages further research that can tackle these computational challenges, including scaling MCMC to large neural networks or developing DP variational inference approaches [41, 45, 70] to the $\beta$D-Bayes posterior.

---

### Official Review · Reviewer_9D13 · 2023-06-30

**Soundness:** 2 fair
**Presentation:** 2 fair
**Contribution:** 2 fair
**Rating:** 6
**Confidence:** 4

**Summary:**

In this paper, authors propose a modified version of one posterior sample (OPS) mechanism. The OPS mechanism releases a sample from a posterior distribution in which the likelihood is tempered based on the privacy guarantee. This can be shown to be a simple instance of exponential mechanism which provides the privacy guarantees. Instead of tempering the likelihood (or equivalently scaling the sum of log-probabilities and taking the exp), authors use the $\beta$-divergence loss as the utility loss for the exponential mechanism. Compared to basic OPS this approach has the benefit that the sensitivities of the log-probabilities don't need to be bounded. Instead using EM with $\beta$-divergence loss, requires the pdf's of the model to be bounded from above (the lower limit is trivially 0). Such a bound exists naturally for almost every distribution (except for something like Dirac's delta dist.), while the log-pdf's typically don't have such bound. Authors further note, that the $\beta$-divergence loss can also be used in other DP Bayesian inference methods. The experimental results demonstrate that the proposed method is able to outperform some previous works in DP logistic regression for various data sets, as well as DP-SGD in learning a simple NN for some $\epsilon$'s and number of observations.

**Strengths:**

The unbounded sensitivity of the log-probabilities is a challenge in all DP probabilistic inference methods. Since the log-probabilities don't typically have a lower bound, the previous approaches typically resort into clipping or otherwise bounding the log-probabilities to obtain bounded sensitivity for the perturbation. The proposed solution overcomes this challenge by using probabilities instead of log-probabilities, thus avoiding any possible clipping bias. Therefore this attempt is an interesting contribution to the DP probabilistic inference.

**Weaknesses:**

The limitations of the method could be further discusses. For example, there is a reason why probabilistic inference is typically implemented using log-probabilities. Using the pdf's in the Bayesian updates would often lead to numerical issues, where as the log-probabilities are typically more stable. Also, since OPS produces only a single sample of the posterior, it could be far away from the posterior mean. And if you would be interested in Bayesian inference, a single sample would not be too interesting as it contains no information on the actual uncertainty.

The experimental results have some clarity issues. For example the lines in Fig 2 seem bit inconsistent between the panes. I will include some questions about these into the questions.

Also, I think comparing the proposed approach to any of the OPS works would be a relevant comparison which is currently missing from the paper.

After rebuttal: Authors have sufficiently addressed my concerns in their rebuttal.

**Questions:**

- Is the convergence of the STAN run for the $\beta$D-Bayes somehow included in the privacy parameters? You do discuss this in Section 4, but I don't see any $\delta$'s reported in the experiments.
- Also, while the STAN gives you the warnings, aren't the tests for the geometric ergodicity there only statistical estimates? Wouldn't this give a further error probability that should be taken into account for the privacy analysis?
- Since you use the pdf's instead of the log-pdf's in the $\beta$D-Bayes, I wonder how will it affect the numerical stability of the inference. Do you need to adjust the precision somehow to compensate this, or will it not be a problem at all? Having some discussion about this would seem appropriate.
- Section 5: You say "Note that [19] still presents the state-of-the-art in logistic regression [40].". Is this really so? I checked the [40] and I cannot see them suggesting this. I would be actually really surprised if the DP-SGD with all the modern accounting machinery would not out-perform the logistic regression method of [19].
- Fig 2: the linestyles in the legend look odd. In the top most figure, the green line is dashed, which is not the case for the two lower ones. Furthermore, the shade of green looks bit different between the first row and the rest. Also, it is bit had to say which gray line is which (the \beta D-Bayes or LogReg (1)). I would encourage the authors to make the line styles in this figure consistent and more distinct from each other.
- Does the PM abbreviation stand for posterior mean in Figs 2 and 3?
- the SGD in Fig 3, is it just non-DP SGD?
- If so, I'm really confused why it performs so poorly in the experiments. Why would you obtain so much smaller loss from the $\beta$D-Bayes approach? I guess both should be optimizing at least similar likelihood function, so the main difference would arise from the prior, which then should have a more limited effect as the number of observations increase.

References are as in the paper:
[19] Kamalika Chaudhuri, Claire Monteleoni, and Anand D Sarwate. Differentially private empirical risk minimization. Journal of Machine Learning Research, 12(3), 2011.
[40] Naoise Holohan, Stefano Braghin, Pól Mac Aonghusa, and Killian Levacher. Diffprivlib: the ibm differential privacy library. arXiv preprint arXiv:1907.02444, 2019.

Minor questions/comments/suggestions:
- Denoting the data set with the same variable as the variable in the integral in (4) looks bit odd. Of course it's technically fine, but you might want to consider changing it.
- line 157: should you have a $-\log f(D; \theta)$ instead of $\log f(D; \theta)$?
- a minor comment: "and the data can always be transformed–without changing the mean estimation–to avoid a very small variance.". Can we really _always_ do this without accessing the data? I would imagine that in order to bound the small variance, you would first need to inspect how small the variance is, and then based on that upscale the data. Now, depending on the size of the data, the variance could be affected by a single neighbouring sample, and hence I would imagine you would need to take this into account in the privacy analysis.
- typo, line 265: "Bayescan"
- typo, appendix, proof for Lemma 1: additional $)$ in the second line


**Limitations:**

There are some limitations that I raised in the weaknesses which should be further discussed.

---

> ### Author Rebuttal · Authors · 2023-08-09
>
> Thank you for acknowledging our contribution to DP probabilistic inference and the potential of moving away from log-score updating. We address your questions and concerns specifically below.
> ### Numerical stability of using pdf's instead of the log-pdf's
> The $\beta$D loss *sums* the p.d.f.s to a power rather than multiplying them (Eq. 3 and 4) and is therefore not subject to numerical issues.
> ### Utility of single sample from the posterior
> We agree that when Bayesian inference is the goal, one sample is of little use. In this work, however, we are not advocating Bayesianism as the correct paradigm for inference, but as a convenient way to release DP parameter estimates as an alternative to noising empirical risk minimizers. One sample from the posterior is an unbiased and consistent estimate of the posterior mean and our experiments show that $\beta$D-Bayes improves estimation performance and extends applicability compared to current methods
>
> If uncertainty quantification is required, the privacy budget can be split and more than one sample can be released from a posterior with higher $\beta$. Interesting further work could look at the utility/privacy trade-off here.
>
> In Sec. 6 ll 369 we will add ‘’Extensions of this work could consider the benefits of dividing the privacy budget to allow for the release of more than one sample paving the way for parameter inference as well as estimation’’.
> ### Distance of single sample from the posterior mean
> Ideally, one would release the posterior mean but randomness of the estimate is required in order to provide differential privacy as per its definition. Methods that noise MLEs (e.g. Chaudhuri [19]) also release one sample that could also be far from the original MLE. Our experiments, e.g. Fig 2 and 3 show that a sample from the $\beta$D-Bayes posterior is closer on average to the data generating parameters than the noised MLE in [19] or a sample from the Gibbs posterior [65] for the same privacy level, so the cost of ensuring privacy is less under $\beta$D Bayes OPS.
> ### Favourable comparison to other OPS works already included
> Fig 2 compares our $\beta$D-Bayes OPS with Minami [65] who do Gibbs posterior OPS and whose method is an improvement of Wang [80] for logistic regression. We show that $\beta$D-Bayes OPS generally improves the performance of Minami [65]. For neural network classification and regression we are not aware of anyOPS methods that apply and therefore we compare to DPSGD instead. We believe one of our key contributions is providing a setting in which OPS can be extended to a wider class of models.
> ### SGD in Fig 3 is non-DP SGD
> SGD is run with the same hyperparameters as DPSGD but without clipping and noising of gradients. We will explicitly mention this.
> ### Explaining the performance of non-DP SGD in the experiments
> The goal of this experiment was the comparison of the private methods. For SGD we chose the same parameters as for DPSGD which includes a small number of epochs. While non-private performance could believably be improved with other hyperparameters, we included SGD as an ablation of DPSGD. We will explain this further in the appendix and Fig 3 no longer compares $\beta$D-Bayes with SGD in the non-private setting.
> ### Utility of stan warnings despite them being statistical estimates
>
> Theorem 1 proves that a sample from the exact posterior is DP, and following Minami et al. [65] and Wang et al. [80], we assume that a sample from a chain after convergence is representative of a sample from the posterior. We use the absence of stan warnings to justify this. While, as you point out, these diagnostics are only estimates, they are widely adopted measures to assess MCMC convergence outside of the privacy setting. It is reasonable to assume that the data holder can choose a large enough number of steps to obtain a posterior sample without the occurrence of such stan warnings
>
> Thank you for pointing out that these diagnostics are missing from the paper. We will report the ESS and R-hat scores in the appendix the number of warm-up steps and iterations used.
>
> We specify on ll 269 how the properties of $\beta$D-Bayes make it suitable for applications of DP-MCMC, where the whole MCMC chain is made private, but we leave investigating this for future work. We will further elaborate on the limitation of stan warnings in Sec. 6.
> ### Current relevance of [19]
> Thank you for checking. [40] is an example use case of [19] (DP logistic regression proposed by Chaudhuri et al.) in 2019 suggesting that [19] was not outdated then. Further, [19] is designed specifically for logistic regression allowing for the use of 2nd order optimisation techniques and privatising the final converged logistic regression parameters. DPSGD on the other hand privatises each gradient step and can only be run for a limited number of iterations, thus introducing more noise for the same level of privacy. We re-formulate this sentence to reflect that “[19] still presents a widely-used implementation of DP logistic regression”
>
> We compare $\beta$D-Bayes OPS with DPSGD in our neural network classification and regression examples (Fig 3 and 6).
>
> ### ‘’[T]he data can always be transformed…to avoid a very small variance."
>
> We agree the original comment was vague, and yes, you do not want to scale by something that depends on the data. A strategy that can always be implemented is to add $0$ mean Gaussian noise with variance $s_0^2$ to the responses $y$. Then you can trivially bound the response variance of the linear regression above $s_0^2$.
>
> To ll 235 we add:
> > In situations where a natural lower bound is not available, one can guarantee this bound by adding iid 0 mean Gaussian noise with variance $s_0^2$ to the observed responses $y$.
>
> This is conceptually different from noising parameter estimate. This repeatedly adds 0 mean noise with a very small variance (we used $s_0^2 = 0.01$) to every observation and doesn’t affect the regression parameter estimates.

---

> > ### Comment · Reviewer_9D13 · 2023-08-15
> >
> > Thank you for your detailed response! Your response addresses most of my concerns and I will raise my score.
> >
> > That being said, there are couple of things I would encourage the authors to include in the final version of this paper.
> > 1. Explicitly mention in your experimental section that the PM stands for the posterior mean, which (I guess) is not DP but serves as the baseline for the proposed method.
> > 2. Explain how the hyperparameters for the DP-SGD comparison were selected. As the paper is not advocating Bayesianism as inference method, it is important that the comparison to the DP-SGD is as fair as possible. If the hyperaparameters are chosen suboptimally, then the DP-SGD might struggle unnecessarily.

---

> > > ### Author Response · Authors · 2023-08-18
> > >
> > > We are pleased our responses address your concerns and thank you for your careful consideration.
> > >
> > > We will ensure the final version of this paper addresses your two points:
> > > 1) We will explicitly mention that the posterior mean (PM) is the point estimate one would ideally release if privacy were not an issue i.e. a non-private baseline.
> > > 2) The DP-SGD parameters were set with hyperparameter tuning to make the baseline as strong as possible. We will explicitly mention this and what the values are in the experimental section.

---

### Official Review · Reviewer_fSk8 · 2023-07-05

**Soundness:** 3 good
**Presentation:** 1 poor
**Contribution:** 2 fair
**Rating:** 5
**Confidence:** 4

**Summary:**

Maintaining differential privacy in a Bayesian setting, can be implemented using Gibbs posterior sampling, which can be viewed as the exponential mechanism with respect to the score function, defined by the the sum of the log prior probability and the log likelihood of the dataset multiplied by a factor governed by the privacy parameter. To ensure differential privacy, this factor must scale with the global sensitivity of the log likelihood, which is unbounded in the general case.

The authors of this paper propose $\beta$D-Bayes, a DP mechanism based on an alternative score function, which is defined by the $\beta$ divergence. This loss function is bounded whenever the underlying distribution is bounded, a much more reasonable assumption. While this method is no longer justified by the theoretical Bayesian framework, it is still consistent under certain assumptions.

While this probability function is intractable in the general case, it can be approximated using MCMC techniques, which are proven to guarantee privacy under additional assumptions. They continue to evaluate numerically and empirically the quality of this new technique.

**Strengths:**

The posterior sampling technique is an important tool in providing private parameter estimations in a Bayesian setting. This techniques has two main drawbacks. The first is the infinite global sensitivity of the score function in the general case, and the second is the intractability of the posterior distribution. This paper provides an important method that reliefs the first issue.

**Weaknesses:**

It might result from my limited knowledge in statistical tools, but I found it hard to parse many of the claims presented in the paper.

Presentation: In many cases, the paper avoids providing full definitions and conditions for the stated claims, and instead refers the reader to other papers or the appendix. The conditions in Theorem 2 and Proposition 2, and the discussion at the first paragraph of section 4, all refer to conditions stated in other papers, and the notation used to state Proposition 1 were defined only in the Appendix, which I found very challenging to follow.

Hard to parse informal claims: While the formal claims were clearly stated, some of the results stated in the introduction are not fully proven, but are based on discussions in sections 3 and 4. While these short discussions might be sufficient for experts in the field, I found it hard to follow. In particular, I did not understand the first paragraph in page 6 which discusses implementation to NNs, and the first paragraph in page 7 which discusses the computational guarantees (as far as I can tell, there are none for $\beta$D-Bayes).

Minor comments:

* In equation 4, the letter $D$ is used twice, once as an argument of the function, and once as the integrated term, which is confusing.

* In Theorem 2, $\theta_{0}^{(\beta)}$ was not defined. Is it possible it is a shorthand for $\theta_{0}^{\ell^{(\beta)}}$?

* In Figure 3, the choice of colors makes it hard to distinguish between $\beta$D-Bayes (PM) and SGD.

=======

**Edit after rebuttal discussion:**

As my main concerns were presentation related, I hope the authors will update their work in accordance with the additional explanations they added in the rebuttal. In particular, in my opinion, the format of the statements should be edited, so that the paper will become self contained.

**Questions:**

As mentioned in the previous section, I had a hard time to parse many of the claims, especially those which were not formally stated. The authors input will be appreciated.

Minor question: In line 52, I failed to understand why the OPS method was presented as an alternative to the sensitivity method, while as the authors explain in later parts, it is actually an implementation of the exponential mechanism with the appropriate sensitivity function?

---

> ### Author Rebuttal · Authors · 2023-08-09
>
> Thank you for acknowledging the importance that moving away from log-score updating contributes to DP probabilistic inference. We address your questions and concerns specifically below.
>
> ### Clarifying Thm. 2 and Prop. 2, the first paragraph of Sec. 4 and Prop. 1
> Space in the paper is limited and we decided to spend what we have explaining the concepts. We propose the following changes to improve this section’s clarity:
> We can discuss the conditions of [64] in relation to the $\beta$D-Bayes posterior before the proofs of Thm. 2 in the appendix.
> Condition 2 required for Prop. 2 is provided in the appendix before the proof of the proposition. We will directly refer to this in the statement of Prop. 2.
> A formal statement of Prop. 12 of Minami [65] and their conditions will also be added to the appendix and correctly referred to in the paper.
> From Prop. 1, $H^{(\beta)}_0$ is the Hessian matrix and $K^{(\beta)}_0$ the cross-product matrix of the gradient vector, we keep their formal definitions in the appendix, but add the following sentence to Prop. 1: “where $K^{(\beta)}_0$ and $H^{(\beta)}_0$ are the gradient cross-product and Hessian matrices for the $\beta$D loss and are defined in appendix Eq. …’’
> We will add a subsection to Appendix A providing formal definitions of all notation.
>
>
> ### Clarifying the first paragraph in page 6 discussing the implementation to NNs
> The first paragraph on page 6 points out that NNs have been shown to outperform logistic regression for modern problems. However, the log-likelihood of a NN classifier is not convex, and therefore the methods of Minami [65] and Chaudhuri [19] cannot be applied. Instead, the state-of-the-art method for DP estimation of NNs is DPSGD which adds noise to minibatch gradient evaluations in SGD and clips the gradients at some value to bound their sensitivity.
> We agree our previous statement on ll 222 was imprecise and we will replace this statement with: “unlike logistic regression, the log-likelihood of a neural network classifier is not convex and therefore the methods of Minami et al. [65] and Chaudhuri (2011) [19] cannot be applied.”
> Further, our description of DPSGD on ll 224 was very brief and relies on the reader to remember DPSGD from ll 46. We therefore replace this with: “DPSGD [1] which adds noise to minibatch gradient evaluations in SGD and clips the gradients at some value to artificially bound their sensitivity.”
>
> ### Clarifying the first paragraph in page 7 which discusses the computational guarantees
> Sec. 4 introduces that while OPS sampling from the $\beta$D-Bayes posterior is $(\epsilon, 0)$-DP, if one uses MCMC to approximate the $\beta$D posterior the MCMC approximation needs to be accounted for in the DP analysis. The cited result from Minami [65] says that if the distribution of the MCMC chain is within in total variation distance $\gamma$ of the $\beta$D-Bayes posterior, then a sample from the MCMC chain is $(\epsilon, \delta^{\prime})$-DP, with $\delta^{\prime} = (1+e^{\epsilon})\gamma$.
>
> We know that if we run MCMC for $N = \infty$ iterations we can achieve $\gamma = 0 \Rightarrow \delta^{\prime} = 0$, but in practice only finitely many iterations are possible. The first paragraph on page 7 evokes the result from Seeman [74] saying that if the chain is geometrically ergodic, then at least order $\log(n)$ ($\Omega(\log(n)$) iterations are required to get a $\delta^{\prime} = (1+e^{\epsilon})\gamma$ to be at most order $1/n$ ($O(1/n)$), a reasonable value for the size of $\delta$ as we state in Line 34. The paragraph then discusses the sampler we consider. We used Stan’s implementation of the No-U-Turn sampler which has been shown to be geometrically ergodic (i.e. fast mixing) and comes with warnings when the chain exhibits evidence of a lack of geometric ergodicity. The conclusion is that running stan for sufficiently many iterations to not receive any warnings provides reasonable confidence of a negligible $\delta^{\prime}$.
>
> However, we agree that our statement on ll 258 can be improved. We will increase the clarity of this passage by
>
> + Defining $\delta^{\prime} = (1+e^{\epsilon})\gamma$ on l 250 and referring to it on l 260
> + Adding to ll 258: ‘’Seeman et al. [74] observed that if the MCMC algorithm is geometrically ergodic achieving a delta smaller than order $1/n$ and preventing data leakage requires the chain to be run for at least order $N = log(n)$ iterations”
> + Finishing the paragraph with: ‘’running stan for a sufficiently many iterations to not receive any warnings provides reasonable confidence of a negligible $\delta^{\prime}$’’
>
> This paragraph does not provide formal guarantees for using stan. Following the works of Minami et al. [65] and Wang et al. [80], we assume that a sample from a chain after convergence is representative of a sample from the posterior, and the paragraph here explains that this assumption is reasonable for the sampler we have chosen.
>
> Clearly formal guarantees are desirable but we believe these are out of the scope of this first paper. We specify on ll 269 how $\beta$D-Bayes is suitable for applications of DP-MCMC, but we leave investigating this for future work. We hope that the existence of a more precise general-purpose DP posterior encourages and facilitates new advances in DP-MCMC.
> ### Why the OPS method is presented as an alternative to the sensitivity method
> The sensitivity method is a subclass of the exponential mechanism that noise-perturbs an estimate with bounded sensitivity. In contrast, OPS (also an instance of the exponential mechanism) samples from a density proportional to a sensitivity-bounded function. We make the distinction as the noise in Bayesian sampling is not artificially added, but naturally present.
> ### Notation in Eq. 4
> Thanks, we have changed this to $\overline{D}$.

---

> > ### Comment · Reviewer_fSk8 · 2023-08-15
> >
> > I thank the authors for the clarification.
> >
> > As most of my concern are presentation related, rather than content related, I still feel like the scientific community will benefit from a revised version of this paper, which better reflects the interesting ideas it discusses.
> >
> > I leave it to the AC to make the call regarding the choice to leave some definition details out of the main body, but in my opinion this is not a good practice despite the space limitations.

---

> > > ### Author Response · Authors · 2023-08-18
> > >
> > > We thank the reviewer for their positive feedback and are pleased our clarifications and proposed revisions were useful. We do not wish our contribution to be undermined by the presentation details kindly pointed out by the reviewer. Should the AC feel appropriate, an alternative to the proposed additions in the rebuttal would be to move Proposition 1 (which is a formal guarantee that has been provided for other OPS methods so far but does not add to the intuition of our method) to Appendix Section A.2.4 where the interested reader can read up on it alongside the accompanying necessary definitions of the Hessian and gradient cross-product matrix. With the space saved, statement and discussion of Assumptions (1) and (2) of Theorem 4 of Miller [64] relevant to Theorem 2 (which we promised in our rebuttal to add to Section A.2.3) can be moved before Theorem 2 on Section 3 ll 196, and our Condition 2 (currently ll 657 in Appendix A.2.5) can be moved before Proposition 2 on ll 283.

---

### Official Review · Reviewer_ADTx · 2023-07-07

**Soundness:** 3 good
**Presentation:** 3 good
**Contribution:** 3 good
**Rating:** 7
**Confidence:** 3

**Summary:**

The paper introduces a privacy mechanism called \betaD-Bayes, which combines the one-posterior sampling (OPS) technique with the \beta-divergence to provide differentially private (DP) parameter estimation for a wide range of inference models. The goal is to ensure that sensitive information in the training data is not leaked when releasing model parameters. The authors extend the applicability of OPS to general prediction tasks and propose \betaD-Bayes as an alternative to the sensitivity method in DP estimation.

The sensitivity method perturbs the function that depends on the sensitive data with noise scaled according to the sensitivity of the function. However, this approach introduces statistical bias and limits the interpretability of the released statistical estimates. OPS, on the other hand, leverages the uncertainty provided by sampling from Bayesian posterior distributions to generate interpretable DP parameter estimates. OPS has been shown to consistently learn the data-generating parameter and is not restricted to specific models like logistic regression.

The authors introduce \betaD-Bayes to make OPS applicable to a broader class of inference models. They combine OPS with a robustified general Bayesian posterior that minimizes the \beta-divergence between the model and the data-generating process. \betaD-Bayes naturally provides a pseudo-log likelihood with bounded sensitivity for popular classification and regression models without modifying the underlying model. This eliminates the need to assume bounded feature spaces or clip statistical functions.

Extensive empirical evidence is provided, including performance comparisons with relevant baselines on multiple datasets and analysis of sensitivity based on sample size and privacy budget.


**Strengths:**

This paper proposes a generalizable approach that can be potentially high impact. I think it could lead to broadening differential privacy research in the Bayesian inference field. The proposed approach is an efficient alternative to the sensitivity methods, and the empirical evaluation shows that the proposed approach outperforms the baselines.

Overall the paper is well-written, the motivation is clear, technical contribution looks correct and strong. It also has a nice related work section and it is easy to understand the contribution.

**Weaknesses:**

The empirical study conducted in Section 5 is limited. I would suggest adding more complex models to the empirical study, e.g. neural network classification and discuss the complexity of such applications.

**Questions:**

Can the proposed method be applied to stochastic MCMC methods?
How it could be extended to neural networks? Does the complexity allow it to be applied to that type of models?

**Limitations:**

This method, as a one posterior sampling (OPS) method, is computationally infeasible since they aim to achieve differential privacy for a sample from the exact posterior distribution. OPS methods rely on using MCMC samples to approximate the posterior distribution. However, ensuring the convergence of the MCMC sampler is crucial to avoid compromising privacy further. This part is missing at the moment, but I understand it is not within the scope of this paper. I think that could be follow-up work.

---

> ### Author Rebuttal · Authors · 2023-08-09
>
> Thank you for acknowledging the potentially high impact that is brought by our paper’s broadening of differential privacy research within Bayesian inference. We address your questions and concerns specifically below.
>
> ### The breadth of our empirical study (incl. neural network classification)
> Thank you for these comments. We included a comparison with logistic regression as this is one of the only models that can currently be tackled by the OPS methods of Wang [80] and Minami [65] as well as Chaudhuri [19], and we wanted to show that even in these cases $\beta$D-Bayes outperformed these approaches.
> However, we believe one of our key contributions is providing a setting in which OPS can be extended to a wider class of models. We illustrated this by considering a regression example (where DP methods struggle outside of conjugate families because of unbounded responses) with a neural network mean function. The results of this are in Fig 3.
> An example of neural network classification was also already included in the original version of our paper, to further demonstrate the flexibility of $\beta$D-Bayes for DP estimation, but sadly the paper is short on space. This appears in Suppl. Fig 6.
> ### Applications of stochastic MCMC methods for neural networks
> The aim of our paper was to establish $\beta$D-Bayes as an alternative to the current Gibbs posterior OPS methods. In the paper, we ran an off-the-shelf MCMC for neural networks with one hidden layer and this appeared to work well In principle, our paper proves that a sample from the exact posterior is DP and from this standpoint as long as the stochastic MCMC is run long enough to convergence then it would be applicable to our paper. A great deal of research has gone into scaling MCMC methods for logistic regression and there are tailored MCMC methods for neural networks as well.
>
> We hope that the considerable improvements in performance make the increased computational costs worthwhile. Our experiments show that $\beta$D-Bayes OPS outperforms Chaudhuri [19] and DPSGD [1] as well as the Gibbs OPS methods. We hope that the performance of our method encourages further research that can tackle these computational challenges, including scaling MCMC to large neural networks or alternatively DP variational Bayes methods to approximate the $\beta$D-Bayes posterior [41, 45, 70]. These however require different theoretical tools that are out of the scope of this paper.
>
> Finally, we note that these procedures cannot be repeated, as repeated estimation would leak information, and only require one posterior sample (after reaching stationarity). As a result, it is reasonable to trade-off performance gains for computational costs.
> To Sec. 6 l. 377 we will add:
> > A further limitation of OPS is the computational burden required to produce posterior samples, particularly in larger neural networks with many parameters. However, we argue that the improved performance justified such a cost and is mitigated by the fact that such inference can only be run once to avoid leaking privacy and that only one posterior sample is required. We hope that our method encourages further research into computational procedures for these models, including scaling MCMC to large neural networks or Developing DP variational inferences approaches [41, 45, 70] to the $\beta$D-Bayes posterior is another option here.
>
> ### Interesting follow up work could look at ensuring the convergence of the MCMC to avoid compromising privacy further
> You are absolutely right. We proved that exact sampling from the $\beta$D-Bayes posterior was $(\epsilon, 0)$-DP where in reality one would normally use MCMC to approximate sampling from this posterior. In Sec. 4 we refer to a Thm. of Minami [65] which says that if the distribution of the MCMC chain is within in total variation distance $\gamma$ of the $\beta$D-Bayes posterior then a sample from the MCMC chain is $(\epsilon, \delta^{\prime})$-DP, with $\delta^{\prime} = (1+e^{\epsilon})\gamma$. The result from Seeman et al. [74] (stated on l. 258) observed that if the MCMC algorithm is geometrically ergodic achieving a delta at most of order $1/n$ and preventing data leakage requires the chain to be run for at least order $N = log(n)$ iterations. Finally, we point out that stan’s implementation of the No-U-Turn sampler has been shown to be geometrically ergodic (i.e. fast mixing) and comes with warnings when the chain exhibits evidence of a lack of geometric ergodicity.
>
> The conclusion is that running stan for sufficiently many iterations to not receive any warnings provides reasonable confidence of a negligible $\delta^{\prime}$.
>
> However, as you point out exact guarantees are crucial to avoid compromising privacy further. We agree it is somewhat beyond the scope of our first paper and an excellent idea for follow-up work.
> There are two avenues here
> 1. Deploy $\beta$D within methods that specifically design MCMC chains to be differentially private (DP-MCMC). Prop. 2 in Sec. 4 outlines the conditions required of the model for $\beta$D-Bayes to be immediately applicable to some popular methods. Incidentally, these methods often rely on stochastic MCMC methods which you mentioned above.
> 2. Try to take advantage of the boundedness of the $\beta$D-Bayes loss function to provide convergence guarantees for a finite sample chain from a particular algorithm. This for example was done in Prop. 13 of Minami [65] for a small class of well-behaved models.
>
> We have alluded to these in Sec. 6 ll 370-377.
>
> We aimed to propose a posterior that is more precise and widely usable than current methods and hope this inspires further research in this direction.

---

> > ### Comment · Reviewer_ADTx · 2023-08-16
> > **Response to rebuttal**
> >
> > Thank you for your detailed response and addressing my comments. I read the other reviews and the answers to the reviews. I already thought the community can benefit from this paper and I believe the revised version with suggested changes will be even stronger. I keep my accept score as is.

---

> > > ### Author Response · Authors · 2023-08-18
> > >
> > > We are really pleased that you feel our proposed changes would further strengthen the paper. Thank you for your feedback and consideration.

---

### Author Rebuttal · Authors · 2023-08-09

We would like to thank all reviewers for their thoughtful feedback and for appreciating the importance of our contribution. We have changed all minor comments, and addressed their suggestions in individual responses.

Alongside our rebuttal we provide a PDF demonstrating how we will alleviate some of the reviewers’ concerns.

**Reviewers 9D13, fSk8 and D3oa** found the demonstration of our results in Fig 2 and 3 confusing. In particular, the plot was cluttered with too many lines; some that looked similar. The goal of Fig 2 and 3 is to compare the performance of the different private methods for the same level of privacy, and not the performance of their non-private analogs. To address these concerns, Fig S1 (in the attached PDF) provides an updated version of Fig 2 where we have removed the non-private methods (grey/black dotted lines). The plot now compares the four DP methods for the same $\epsilon$ on simulated and real data sets and shows that for large enough \# observations $\beta$D-Bayes OPS incurs the least statistical error. We will also remove the non-DP methods from Fig 3 to improve its readability.

Following comments from **Reviewer D3oa**, we also extrapolated the figures to display the full data set sizes (to the benefit of our method). Before we presented the results only on a subset of the data for clarity reasons. Now that the dashed lines are removed, the results on the full data set can be displayed without loss in readability. We run the method on repeated subsamples of the data in order to provide an idea of how the methods compare for different sample sizes. We do not consider subsamples for computational reasons. We could run our simulations for more than $n = 1000$, however from the top line of Fig S1 (in the attached PDF), we believe it is already clear that $\beta$D-Bayes performs the best. We hope that from the updated Fig S1, it is clear that for the simulations and real data for all considered values of $\epsilon$ for large enough \# observation $\beta$D-Bayes is the best (lowest for log RMSE top plots, highest for AUC bottom plots).

Fig S2 (in the attached PDF) illustrates a separate comparison of private $\beta$D-Bayes inference with its non-private analog (i.e. the posterior mean) and these plots for all the methods will be added to the appendix. As pointed out by **Reviewers 9D13 and D3oa**, PM stands for posterior mean and this is now explicitly noted in a revised version of the paper.

Fig S3 (in the attached PDF) addresses the comment of **Reviewer D3oa** who asked if $\beta$D-Bayes could ‘’replicate[d] a finding without DP’’. For logistic regression, we understood a finding as estimating a coefficient as being of a particular sign. For our real and simulated data, Fig S3 looks at the CSA (correct sign accuracy), the proportion of the time the DP-parameter estimates from the different methods agree with those of a non-private baselines (i.e. l2-penalised logistic regression implemented in sklearn). The plot demonstrates that the estimates from $\beta$D-Bayes OPS would agree with the sign of the non-private method more of the time than the other methods.

---

### Decision · Program_Chairs · 2023-09-21

**Decision:**

Accept (poster)

**Comment:**

The paper presents a novel method for differentially private Bayesian inference. After author rebuttal and discussion, all reviewers recommend acceptance.

For the final version of the paper, the authors should update the presentation to make the paper better self-contained. Especially the theorem statements should be fully self-contained and not rely on references to other papers.